# Competition-driven eco-evolutionary feedback reshapes bacteriophage lambda's fitness landscape and enables speciation

Michael B. Doud [1,2], Animesh Gupta [2], Victor Li[2], Sarah J. Medina[2], Caesar A. De La Fuente [2] & Justin R. Meyer [2] ✉

A major challenge in evolutionary biology is explaining how populations navigate rugged fitness landscapes without getting trapped on local optima. One idea illustrated by adaptive dynamics theory is that as populations adapt, their newly enhanced capacities to exploit resources alter fitness payoffs and restructure the landscape in ways that promote speciation by opening new adaptive pathways. While there have been indirect tests of this theory, to our knowledge none have measured how fitness landscapes deform during adaptation, or test whether these shifts promote diversification. Here, we achieve this by studying bacteriophage $\lambda$, a virus that readily speciates into co-existing receptor specialists under controlled laboratory conditions. We use a high-throughput gene editing-phenotyping technology to measure $\lambda$'s fitness landscape in the presence of different evolved-$\lambda$ competitors and find that the fitness effects of individual mutations, and their epistatic interactions, depend on the competitor. Using these empirical data, we simulate $\lambda$'s evolution on an unchanging landscape and one that recapitulates how the landscape deforms during evolution. $\lambda$ heterogeneity only evolves in the shifting landscape regime. This study provides a test of adaptive dynamics, and, more broadly, shows how fitness landscapes dynamically change during adaptation, potentiating phenomena like speciation by opening new adaptive pathways.

Fitness landscapes were first conceived of by Sewall Wright as a conceptual framework for visualizing the connection between genotype and fitness[1]. Conceptually, fitness landscapes are often projected as two-dimensional distributions of fitness peaks and valleys, and mutations allow populations to move randomly across the topography. If mutations reposition an individual higher up a fitness peak, survival becomes more likely, and conversely survival is less likely during descent into fitness valleys. As populations diverge to climb distinct fitness peaks and become separated by valleys, this process can result in further diversification, and eventually speciation. Although compelling and intuitive when presented as static, three-dimensional distributions, fitness landscapes are far more complex due to the high

dimensionality of genotype space, non-additive interactions between multiple mutations (epistasis) that distort landscape topography, and various biotic and abiotic forces that dynamically reshape fitness landscapes[2].

Adaptive dynamics theory (ADT) is a framework for studying evolutionary change in a setting where the fitness of individuals is not static over time, but can be affected by the frequency of the individuals as well as changes in their ecology[3–5]. ADT is an extension of game theory[4,6] and has been applied to study evolution of resource-limited populations, where competition can dynamically influence fitness landscapes. One of the theory's key predictions is that as populations adapt and compete, there are resulting deformations in their fitness

[1]Department of Medicine, Division of Infectious Diseases and Global Public Health, University of California San Diego, San Diego, CA, USA. [2]Department of Ecology, Behavior and Evolution, University of California San Diego, San Diego, CA, USA. ✉e-mail: jrmeyer@ucsd.edu

landscapes that promote diversification and speciation. For example, under a simple model of evolution with a fixed fitness landscape, populations evolve towards fitness peaks and can become trapped on local fitness optima. In contrast, ADT predicts that when a population adapts to fill a specific niche, increased competition for limited niche resources will decrease the fitness payoff of the niche, thereby shrinking the associated peak. When this peak is lowered, the population is released to explore other regions of the fitness landscape. ADT predicts that this eco-evolutionary feedback can deform the fitness landscape to allow new populations to ascend new fitness peaks. The original population remains well-adapted to the original peak as the new population emerges, allowing for ecological diversification and speciation.

There is limited experimental evidence supporting the ADT model. Bono et al. used experimental evolution of a bacteriophage to show that the rate of acquiring the ability to infect a new host was associated with the degree of competition between viruses to infect host cells[7]. While this study experimentally demonstrated the role that competition can play in facilitating the emergence of new virus phenotypes, it was unclear whether changes in the fitness landscape during adaptation facilitated this ability to infect a new host. Spencer et al. used experimental evolution of bacteria that diversified metabolic

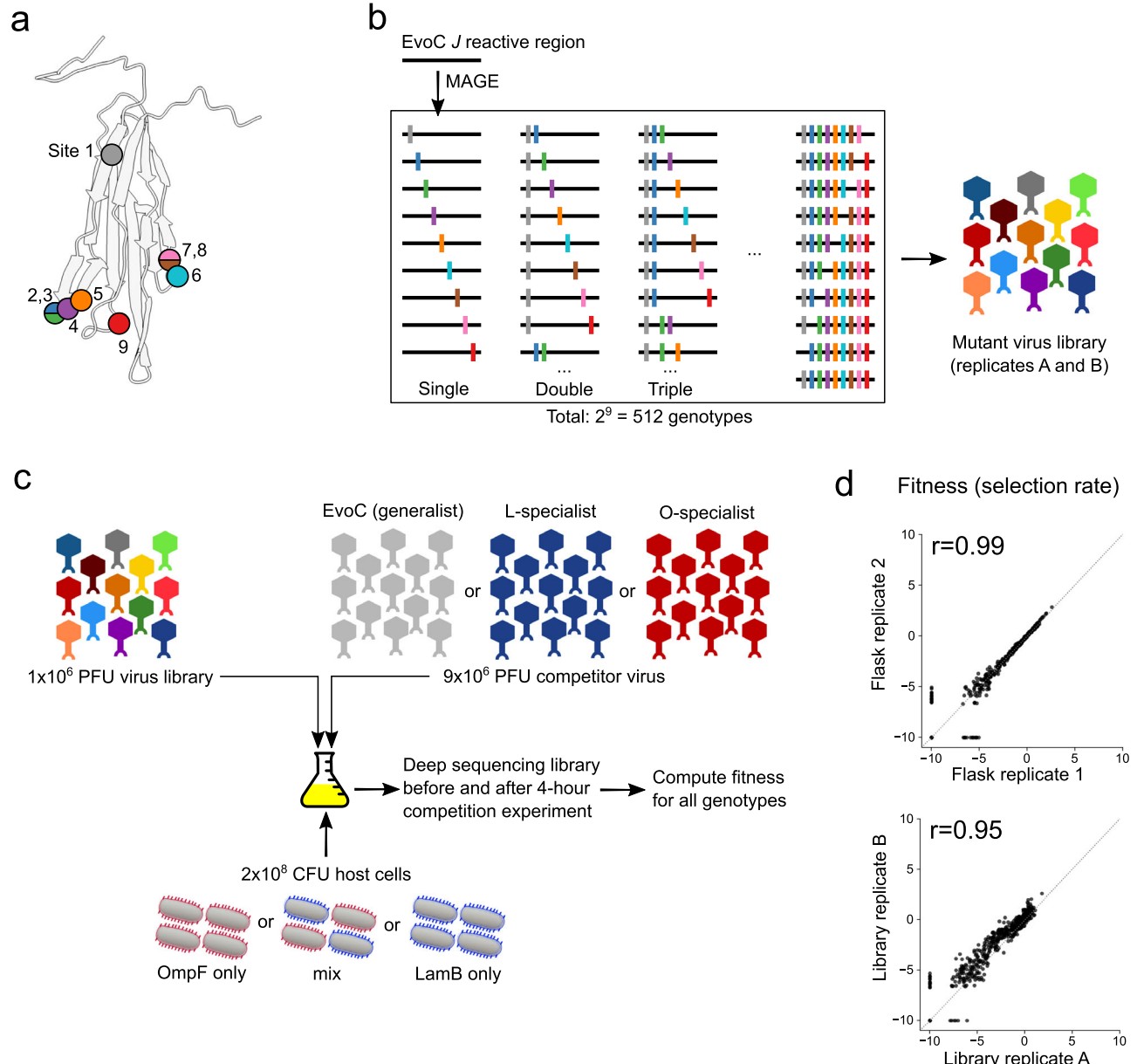

**Fig. 1 | Overview of experimental measurements of fitness landscapes.**
**a** Mutations at nine nucleotide sites in the *J* gene were observed during longitudinal population sequencing of *λ* while evolving from a generalist into two specialists (also Supplementary Fig. 1). Some mutations are in the same codon: mutations 2 and 3 (in isolation or in combination) introduce the same nonsynonymous change at that codon; mutations 7 and 8 (and their combination) introduce distinct nonsynonymous changes in that codon. The amino acid sites associated with the nine nucleotide mutation sites are shown as colored circles on the predicted protein structure of the distal portion of the J protein. **b** Nine mutations were introduced combinatorially by MAGE into the generalist EvoC *J* sequence and subsequently incorporated into a mutant virus library of *λ* particles carrying all viable genotypes. **c** Mutant *λ* virus libraries were mixed with a competitor *λ* strain and cultured for four hours (approximately two infection cycles) to allow for viruses to compete for host cells expressing only OmpF, only LamB, or an even mixture of these cells. The fitness of each genotype was computed using genotype frequencies measured by deep sequencing before and after the competition experiment. **d**, Measurements of fitness were highly correlated between experimental replicate competition flasks and between independently generated mutant virus libraries. Representative samples are shown here, comprehensive results are shown in Supplementary Fig. 3.

strategies for carbon utilization to show that the likelihood of diversification increased only after the population adapted to an initial environment[8]. While this is in line with ADT, this result could also be explained by historical contingency where mutations early in adaptation potentiate future adaptations through genetic interactions[9].

What remains missing to our knowledge as evidence for the ADT model are direct measurements of how the fitness landscape deforms during adaptation and a test of whether these deformations promote diversification. Many studies have shown that shifts in abiotic conditions[10–13] and biotic conditions[14–17] can deform landscapes to open new evolutionary potentials. However, the general extent to which landscapes are deformed as a product of resource competition, and the resulting implications on adaptation and diversification, remain uncharacterized[16]. Here we achieve these missing elements by measuring the fitness landscape of a virus under different competitive contexts and show that the landscape significantly deforms as new competitors evolve, and that these deformations set the stage for diversification.

For this work, we focused on bacteriophage $\lambda$, which was previously shown to speciate into two reproductively isolated receptor specialists within 280 hours of laboratory culturing[18]. When a generalist $\lambda$ (EvoC) that can use two receptors, LamB and OmpF[19], is cocultured with a mixture of two hosts that vary in whether they express LamB or OmpF, EvoC reproducibly evolved into two receptor specialists[18]. Diversity evolved through mutations in the host-recognition gene $J$ which modulated host receptor preferences. Here, we measured fitness landscapes of the $J$ mutations that evolve during speciation using a high-throughput gene editing-phenotyping technology (MAGE-Seq)[17,20], and used simulations of $\lambda$ evolution to explore how adaptation along these fitness landscapes can lead to diversification. Our goals were two-fold: first, to characterize properties of the fitness landscape, such as epistasis and pleiotropy, that contribute to both genetic constraints and opportunities during the evolution of host-receptor tropism and viral speciation. Our second goal was to test a key prediction made by ADT that eco-evolutionary feedback leads to deformations in the fitness landscape which promote diversification. We find that $\lambda$'s fitness landscape significantly deforms as new competitors evolve, and that these deformations set the stage for diversification.

## Results and discussion

### Evolutionary replay of EvoC speciation

The previously published $\lambda$ speciation experiment was conducted over a decade ago in a different laboratory[18]. We repeated the experiment to ensure repeatability in a new location and to preserve samples every 40 hours for longitudinal analyses. We then sequenced the $J$ gene from two specialists isolated at the endpoint (a LamB-specialist and an OmpF-specialist). Next, we deep sequenced $J$ from the population of viruses preserved at each time point to monitor $J$'s molecular evolution. This allowed us to observe the rise of the mutations found in the specialists and to track other contending mutations not present in the endpoint specialist genomes. The amplicon sequencing revealed the appearance of nine $J$ mutations over the entire experiment (Fig. 1a), five of which were observed in the endpoint specialists (Supplementary Fig. 1). Each of the nine mutations resulted in nonsynonymous changes to a region of the J protein known to interact with the host receptor[21] and where previous adaptive mutations have been observed to evolve in the laboratory and nature[22].

### Reproducible measurements of the fitness landscape that leads to speciation

Measuring the fitness landscape for a gene the size of $J$ (3399 nucleotides) is experimentally intractable due to the astronomical number of possible genotypes generated by combinatorics. Given this, we focused on measuring the landscape of all possible combinations of the 9 mutations observed ($2^9 = 512$ $J$ alleles) in the evolutionary replay experiment. Using a 'one-pot' approach, we engineered all 512 alleles into a heat-inducible lysogenic $\lambda$ strain two separate times (Fig. 1b). The resulting lysogen libraries were sequenced, confirming full allelic representation at relatively uniform frequencies (Supplementary Table 1, Supplementary Fig. 2). These lysogen libraries were then induced to produce infectious $\lambda$ virus libraries which were used to measure fitness across a variety of environments.

We set out to measure the fitness of each genotype in parallel by competing the library phages and monitoring changes in genotype frequencies through deep sequencing (Fig. 1c). We hypothesized that the structure of the fitness landscape would depend on emerging receptor specialist phenotypes: for example, if an OmpF-specialist becomes common, $\lambda$ genotypes specialized on LamB should experience less competition to infect hosts expressing their preferred receptor. Concurrently, the competition between OmpF-specialists should become more intense for their preferred resource (OmpF-expressing host cells) as they deplete the host population. Given this, we designed competition experiments to measure the fitness landscapes in five environments spanning the spectrum of receptor competition (Fig. 1c). Three treatments were set up similarly to the evolution experiment, with an equal mixture host cells expressing either LamB or OmpF, and the treatments varied by spiking in an excess (9:1) of different $\lambda$ competitors: the initial generalist ancestor (EvoC), the endpoint L-specialist, or the endpoint O-specialist. Two additional treatments were also tested that represent the extremes of the receptor competition spectrum, where only one host cell type was present (either OmpF- or LamB-expressing host cells) and in both cases EvoC was used as the competitor strain. In all cases, the virus mixture was added to host cells at a low multiplicity of infection (0.05) to limit co-infection, and the cultures were incubated for four hours in shaking flasks at 37°C, allowing for roughly two cycles of $\lambda$ infection.

Deep sequencing was used to measure the frequencies of all genotypes in the libraries before and after the four-hour competition by isolating phage genomic DNA and using PCR to add unique molecular barcodes and Illumina sequencing adaptors. The competitor viruses in all cases were excluded from PCR amplification by introducing synonymous mutations within the primer binding sites used to generate sequencing libraries. Sequencing errors were resolved by building consensus sequences for each unique molecular barcode (representing a unique observation of a genotype from the competition experiment), and genotype frequencies were calculated based on the number of unique molecular barcodes observed in association with each genotype. Competition samples were sequenced to a median effective depth of $1.4 \times 10^6$ unique molecular barcodes per sample (range: $1.0 \times 10^6$ to $1.5 \times 10^6$, Supplementary Table 1). To compute a fitness value for each genotype in the library, we calculated the selection rate relative to the ancestral EvoC genotype. Selection rate is defined as the logarithm of the fold change in a genotype's frequency relative to that of a reference genotype, over a unit of time, which in our experiments was 4 hours. The ancestral generalist sequence EvoC is used as the reference genotype in all analyses, so the reported fitness values are always relative to EvoC. Our method provided highly reproducible measurements of genotype fitness. We measured fitness using two independently generated mutant virus libraries, and each library was measured in the five environments in triplicate competition flasks. The resulting fitness measurements were highly correlated, both between replicate competition flasks (Pearson correlation coefficient 0.96−0.99) and between independent virus libraries (Pearson correlation coefficient 0.95−0.98) (Fig. 1d, Supplementary Fig. 3).

### Competitor-dependent epistasis reshapes fitness landscapes

It remains an open question to what degree environmental change (such as the spectrum of competitive environments we used to measure fitness landscapes) can reshape fitness landscapes through

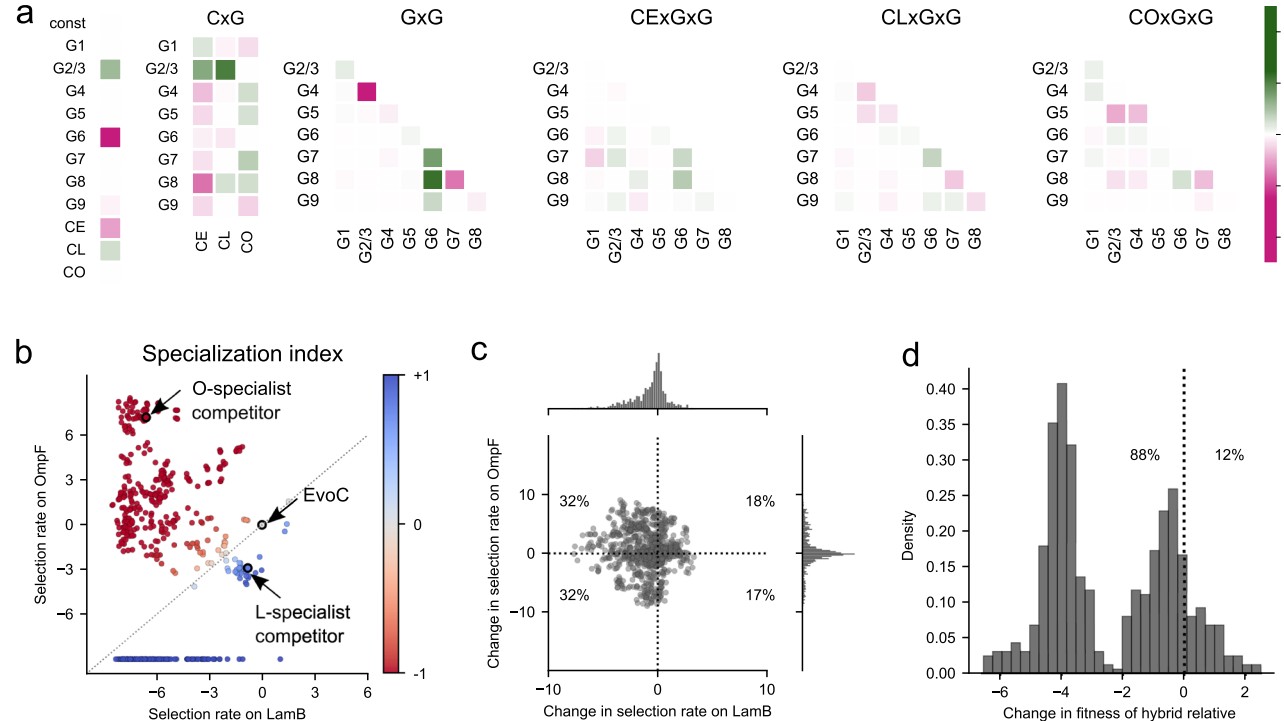

**Fig. 2 | Statistical properties of the J fitness landscape. a** Heatmap of coefficients for a linear regression model predicting virus fitness from genotype and competitor context after LASSO regularization. Fitness is predicted from individual mutations ("G" terms), competitor virus context ("C" terms: CE = EvoC generalist competitor, CL = LamB specialist competitor, CO = OmpF specialist competitor), and non-additive interaction terms including competitor-by-mutation ("CxG"), mutation-by-mutation ("GxG"), and competitor-by-mutation-by-mutation ("CxGxG"). Larger coefficients denote stronger influence on fitness predictions. Mutations 2 and 3, alone and in combination, introduce the same amino-acid change and are highly correlated in their effects on fitness (Supplementary Fig. 6 and 7), thus are treated as a single term. Non-zero coefficients for CxG and CxGxG terms indicate that the competitor context contributes to changes in the fitness landscape by modifying both the effects of individual mutations and the pairwise epistasis between mutations. **b** A specialization index (SI) is calculated for each genotype based on the fitness measured on each host receptor in isolation. SI ranges from -1 (OmpF specialist) to +1 (LamB specialist) (see Methods). By definition, EvoC has SI = 0 since all fitness measurements are in reference to this genotype. Each point represents a genotype from the combinatorial library and is colored by SI. Many genotypes have relatively high SI, which contributes to changes in fitness landscapes when varying host receptors are available for infection. Unobserved genotypes in a given condition are assigned a selection rate of −10 for visualization purposes. **c** Pleiotropic effects of mutations on fitness for each receptor. From any starting genotype in the library, most point mutations reduce fitness on LamB (the native receptor), but the change in fitness on OmpF is evenly distributed around zero. **d** Hybridization between specialists tends to incur a fitness cost. Genotypes with the highest absolute specialization indices (27 O-specialists and 27 L-specialists, see Supplementary Fig. 4) were used to generate hybrids in silico that contain combinations of mutations from the parental specialists. On average, the hybrid genotypes were less fit than the average of the parental specialists.

epistasis[2]. Since we measured fitness landscapes in the presence of three different competitors, we were able to quantify competitor-dependent effects on the landscape. We used multiple linear regression to model fitness as a function of the nine phage mutations and the competitor. We also included interaction terms in the model for competitor-by-mutation (CxG) and competitor-by-mutation-by-mutation (CxGxG) effects, to allow for the possibility that the fitness effects of some mutations, and epistasis between pairs of mutations, could also depend on the competitor. This approach allowed us to infer the joint influence of mutations and the competitor on the structure of the fitness landscape. We used LASSO regularization to reduce the model complexity to avoid over-fitting, and the resulting coefficients on the predictor variables are shown as a heatmap in Fig. 2a, where the most positive and negative coefficients denote the strongest impacts on predicted fitness. Some mutation terms (G) and pairwise epistasis terms (GxG) are strongly predictive regardless of competitor, however, there are a few strong competitor-dependent effects on individual mutations (CxG), and to a lesser degree many competitor-dependent pairwise epistasis terms (CxGxG). The finding that epistatic interactions between mutations can vary across competitive contexts is in line with broader work demonstrating more generally how environmental perturbations can lead to shifts in fitness landscapes through epistasis, such as the effects of magnesium ion concentration

on ribozyme activity[23], nutritional resources for bacterial growth[10,12], host species for viral infection[15,24], and inducer concentrations regulating a synthetic gene regulatory network[13]. Our results suggest that the competitor virus phenotype contributes to changes in the fitness landscape, not only by influencing the fitness effects of individual mutations, but also by influencing pairwise epistasis between mutations. More generally this suggests that as a new phenotype evolves in the population, it has the potential to deform the fitness landscape – potentially opening new adaptive pathways and ecological opportunities.

**In silico phenotyping reveals a spectrum of generalist and specialist phenotypes**

We next examined our results through the lens of receptor-use phenotypes by phenotyping in silico all genotypes in the library. Broadly speaking, receptor 'generalists' are defined to utilize more than one receptor, while 'specialists' predominantly utilize a single receptor. Based on competition experiments in which only LamB or OmpF-expressing host cells were provided for infection (Fig. 1c), we derived each genotype's fitness on each of the individual receptors OmpF and LamB. We then calculated a specialization index (SI) for every genotype (Fig. 2b, Methods). Genotypes that are much more fit on OmpF than LamB are OmpF specialists and have an SI approaching −1;

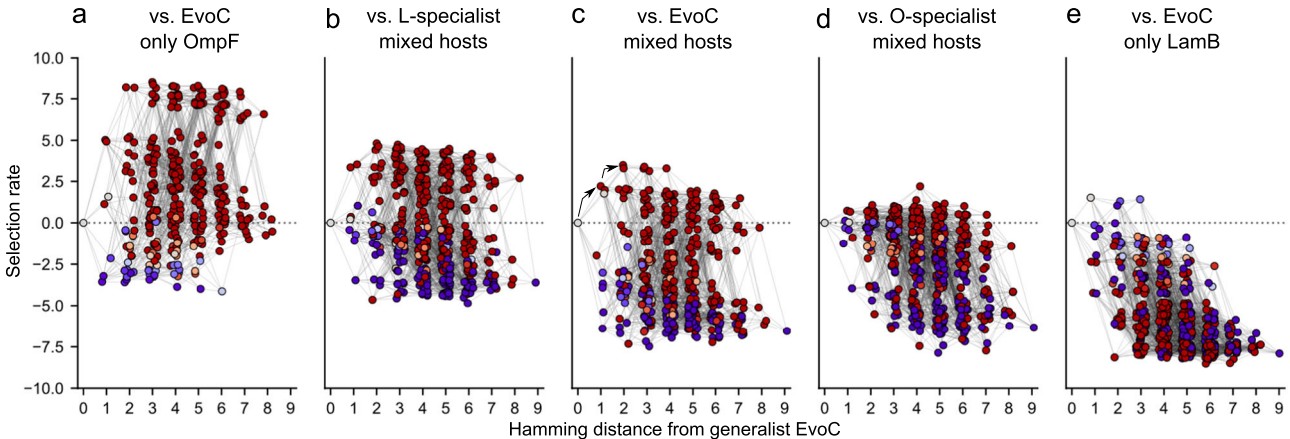

**Fig. 3 | Fitness landscapes are reshaped by competition between viruses for host cells.** Fitness landscapes are represented by hypercubes where each point denotes a genotype. Points are connected by a line if they differ by only a single mutation. The points are separated along the x-axis by hamming distance from EvoC (i.e., number of mutations), and fitness (selection rate) is plotted on the y-axis. Some genotypes in some conditions are not shown because their fitness was so low that they were not observed at the end of the competition experiment and selection rate could not be quantified. Genotypes are shaded according to specialization index as in Fig. 2b (red = OmpF specialist; blue = LamB specialist, grey = generalist). Lines connect two genotypes if they are separated by a single mutation. **a** competition against EvoC for exclusively OmpF-expressing host cells; **b** competition against L-specialist for mixture of host cells; **c** competition against EvoC for mixture of host cells; two black arrows represent sequential mutations that lead from the ancestral sequence to a fitness peak defined by O-specialists; **d** competition against O-specialist for mixture of host cells; **e** competition against EvoC for exclusively LamB-expressing host cells. See Supplementary Fig. 15 for another visualization of this figure showing the individual points.

conversely, LamB specialists have an SI approaching +1, while genotypes agnostic to receptor have a SI near 0. As expected, the evolutionary replay experiment end-point L-specialist and O-specialist genotypes (that were later used as the competitor viruses in the fitness landscape measurements) had SI measurements confirming their respective specializations (Fig. 2b). Most genotypes in the virus library were strong LamB or OmpF specialists, including many LamB specialists that were completely unable to infect through OmpF. This is consistent with prior work suggesting that mutations allowing a generalist to specialize on one host are generally accompanied by a trade-off in fitness on other hosts[25].

### Effects of mutation and hybridization on receptor specialization phenotypes

We next analyzed how mutation and hybridization between genotypes impact receptor specialization phenotypes. Acquisition of any of the 9 mutations, on average (across all genotype backgrounds in the library), tend to be deleterious to LamB fitness and have a relatively equal chance of being beneficial or deleterious to OmpF fitness (Fig. 2c). We interpret this to mean that $\lambda$ was already well-adapted to use its native receptor, LamB, and there are fewer mutations available to improve on it. In contrast, the use of OmpF as a second receptor is a novel function EvoC gained through experimental evolution[19], and there remain ample opportunities to gain or lose fitness through additional mutations. We were particularly interested in whether genomic incompatibilities between specialists, which we had previously observed in a single pair of specialists[18], might contribute more generally to speciation of diverging specialists. We examined a set of the most specialized OmpF and LamB specialists (Supplementary Fig. 4) and asked whether hybrids derived from two specialists became more or less fit than the parental specialists. Most hybrids (88%) were less fit than the average of the parental specialists (Fig. 2d), congruent with our prior results with a single pair of specialist genotypes[18]. This is likely an underestimate of the fitness loss that occurs during the hybridization of opposing specialists, because it does not include cases of hybrid genotypes that completely lost the ability to infect through OmpF, which could not be analyzed quantitatively because their fitness on OmpF is undefined. Overall, the widespread observation of genetic incompatibilities between mutations observed in

different receptor specialists shows that Mueller-Dobzanski incompatibilities[26], which have not been extensively studied in viral speciation, can contribute to $\lambda$'s speciation, in addition to the reproductive isolation ensuing from the preference to infect different host cells[27,28].

### LamB and OmpF specialists occupy distinct fitness peaks in diverging environments

Fitness landscapes across the five competition conditions are shown in Fig. 3. We visualized the landscapes by plotting the fitness (selection rate) of all genotypes, arranged by the number of mutations acquired relative to EvoC, and colored by SI, from red (strong O-specialist) to grey (generalist) to blue (strong L-specialist). The landscapes are arrayed from left to right based on the gradient of strength of competition for LamB versus OmpF. As expected, O-specialists have the highest fitness in the landscapes where there is strong competition for LamB (for example, as in Fig. 3b in the presence of an L-specialist competitor) and most L-specialists are maladapted and have lower fitness than the EvoC ancestor. O-specialists also have high fitness in the landscape where both hosts are available and the competitor is a generalist (Fig. 3c), which might seem counterintuitive, although it can be explained by the previous observation that EvoC has a slight preference for LamB[18], effectively acting like a weak L-specialist competitor. In the flasks where both hosts are available and the O-specialist competitor makes up 90% of the viruses (Fig. 3d), the L-specialists gain fitness, the O-specialists lose fitness, and some L-specialists out-compete the EvoC ancestor. This trend continues in the condition where only LamB cells are available (Fig. 3e). This shift from selection favoring OmpF-specialists to LamB-specialists is in line with predictions by ADT that as competition for one niche intensifies, selection will favor genotypes able to access alternative niches.

Overall, these fitness landscapes visually demonstrate that when different competitor virus phenotypes emerge in the virus population, the resulting changes in resource competition sculpt the fitness landscapes to favor different phenotypes in different conditions. To our knowledge this is the first measurement of the effect that competitor viruses have on reshaping fitness landscapes and the positions of generalists and specialists on different fitness peaks. Martin et al. observed complex fitness landscapes of *Cyprinodon* pupfishes, where a

a

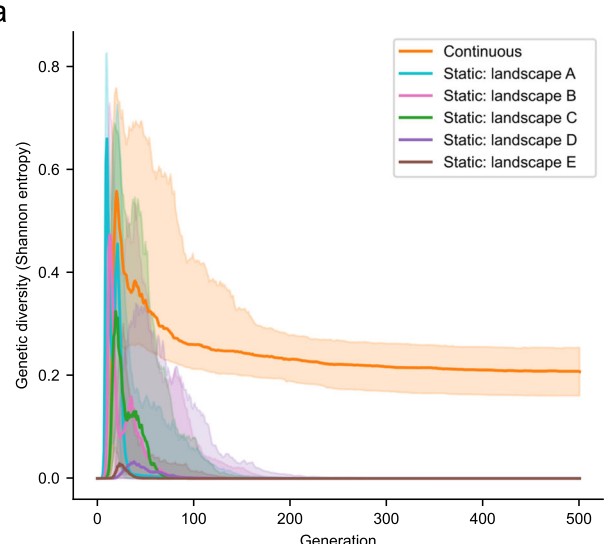

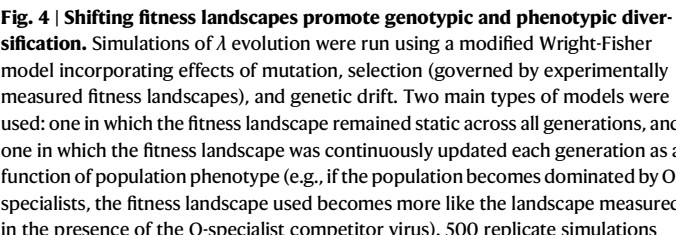

b

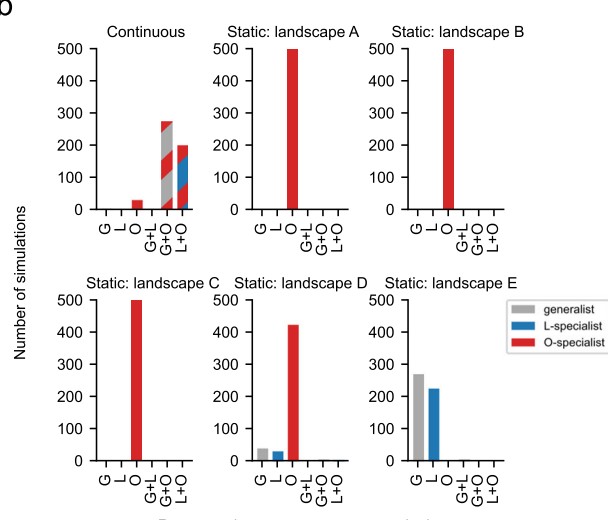

**Fig. 4 | Shifting fitness landscapes promote genotypic and phenotypic diversification.** Simulations of $\lambda$ evolution were run using a modified Wright-Fisher model incorporating effects of mutation, selection (governed by experimentally measured fitness landscapes), and genetic drift. Two main types of models were used: one in which the fitness landscape remained static across all generations, and one in which the fitness landscape was continuously updated each generation as a function of population phenotype (e.g., if the population becomes dominated by O-specialists, the fitness landscape used becomes more like the landscape measured in the presence of the O-specialist competitor virus). 500 replicate simulations were performed for each model. The static landscape labels A-E correspond to the landscape labels in Fig. 3. **a** Genetic diversity was computed at each generation. Solid lines represent the median and shaded regions represent the interquartile range. Only the shifting landscape model results in genetic diversity. **b** Phenotypic diversity computed at the endpoint of each simulation. The number of simulations arriving at the designated combination of phenotypes (>=2.5% abundance) is plotted. Only the shifting model results in combinations of two coexisting receptor phenotypes. Examples of individual simulations under each model are shown in Supplementary Fig. 5.

generalist species occupied a local fitness optimum separated by fitness valleys from even higher fitness peaks containing specialists[29]. They hypothesized competition may allow for the generalists to 'escape' from being trapped on the generalist peak, although intriguing follow-up work found that these fitness peaks were surprisingly static and independent of competitor frequency[30]. This is in contrast to our findings with $\lambda$ where there are clearly competitor-dependent shifts in specialist fitness peaks, which are generally accessible to the generalist without crossing a fitness valley. The differing results in these studies on *Cyprinodon* and $\lambda$ might be due to any number of factors, including differences in the biology and ecology between pupfish and their prey in natural environments, and viruses and their prey in controlled laboratory environments, as well as differences in methodology in measuring fitness landscapes between these studies.

### Dynamic fitness landscapes promote genotypic and phenotypic diversification

We next asked whether the structure of these fitness landscapes could help explain the process by which a receptor generalist could evolve into two coexisting receptor specialists, as we have seen reproducibly in laboratory evolution[18]. Visual inspection of the fitness landscape corresponding to a population of the ancestor generalist virus without any specialist competitor present (Fig. 3c) shows that an early fitness peak can be reached by an adaptive walk taking several mutations (black arrows), and that this peak leads to specialization on OmpF. We envisioned that as a virus population evolves by ascending this peak, and as the population phenotype shifts closer to becoming dominated by O-specialists (corresponding to the experimental conditions used to measure the fitness landscape in Fig. 3d), that the narrowing fitness gap between O-specialists and L-specialists might allow for L-specialists to emerge. Furthermore, it appears from visual inspection that as the population becomes dominated by exclusively OmpF-utilizing viruses (and in the limiting case, removing all available OmpF-expressing cells, represented by the experimental conditions in Fig. 3e) that the resulting fitness improvement of L-specialists could permit their emergence and

co-existence with O-specialists. More generally, in line with ADT, we hypothesized that shifts in resource use (receptor usage) will alter the fitness landscape structure in ways that promote diversification.

We tested this hypothesis using computer simulations of $\lambda$ evolution, using our experimentally measured fitness landscapes to assign reproductive success probabilities to genotypes. We used a modified Wright-Fisher model to simulate the effects of mutation, selection, and genetic drift on $\lambda$ evolution. We used two general types of models for simulating evolution: one using a single static fitness landscape throughout the simulation, and one using a shifting fitness landscape. The static landscape models simulated evolution on each of the five landscapes separately, using a constant landscape throughout each simulation. The shifting models used experimentally measured fitness landscapes from all five environments, placed along a "landscape axis" with coordinates defined by population phenotype (i.e., the average specialization index (SI) of the population, see Methods and Supplementary Fig. 10). For each evolved generation in the shifting models, the fitness landscape used to govern selection on that generation was re-assessed based on the simulated population's average SI. Among the two shifting models we implemented, a fitness landscape was either computed as a weighted average of the nearest two empirical landscapes (continuous shifting model), or the nearest empirical landscape was used without interpolation (discrete shiting model). In this manner, the fitness landscapes underlying selection in the shifting models fluctuate in response to the emergence of new phenotypes.

We performed 500 replicate simulations for each of the evolutionary models. All simulations started with a homogenous population of the ancestor EvoC genotype, and each simulation progressed with population sizes equivalent to those observed during the evolution (5 ×10[9] viral particles) for 500 generations (Supplementary Fig. 5). Figure 4a shows the average genetic diversity plotted across generations under each model. Although in early generations most models had similar average levels of genetic diversity as new genotypes sweep through the population, diversity is only maintained in the shifting landscape model. While the static models tend to converge on single

genotypes dominating the population, only the shifting model allows for the prolonged co-existence of multiple genotypes.

We hypothesized that the increased genetic diversity under the shifting model was reflective of phenotypic diversity in receptor specialization. We used in silico phenotyping (Fig. 2b) to ascribe receptor preferences to the viruses surviving at the end of the simulations (Fig. 4b). Most simulations under the shifting model resulted in more than one virus phenotype (an O-specialist co-existing with either an L-specialist or a generalist). In contrast, each of the static models nearly always arrived at a viral population with a single phenotype. The static landscape 'C' that best reflects the initial conditions of the evolution experiment only favors the emergence of a single phenotype (O-specialist). Only by allowing for the eco-evolutionary feedback imposed by emergent viral phenotypes reshaping the fitness landscape during $\lambda$ evolution is it possible for co-existing specialist phenotypes to evolve, leading to speciation. Our results build upon existing work detailing the effect of eco-evolutionary feedback on adaptive evolution. These results that we obtained using empirical fitness landscapes are in striking agreement with a recent study using theoretical fitness landscapes, where it was also found that dynamic fitness landscapes influenced by eco-evolutionary feedback promoted genetic diversity, whereas static fitness landscapes only allowed for one dominant genotype[31]. Other work examining the deformability of empirical landscapes by metabolic mutations in *E. coli* has suggested that static fitness landscapes can be used to forecast evolution over short evolutionary distances, and only under longer mutational distances do the shifts in fitness landscapes become meaningful[32], however, our work presents an example where rapid eco-evolutionary feedback can significantly reshape landscapes in a short amount of evolutionary time.

In summary, we find that the structure of $\lambda$'s fitness landscape is dependent upon environmental factors including host cell availability and competing virus phenotypes. The fitness effects of individual mutations, as well as epistatic interactions between mutations, are modulated by biotic changes in the environment when competing viruses with different receptor preferences alter resource availability. Furthermore, we find through computer simulations of $\lambda$ evolution that a shifting fitness landscape, reshaped through epistasis and pleiotropy as new population phenotypes emerge to compete for new niches, is necessary for the outcome of viral speciation as we have seen experimentally[18]. These results provide the first experimental demonstration of ADT by directly observing deformations in a fitness landscape across a spectrum of carefully controlled laboratory environments designed to perturb the nature of virus-virus competition during $\lambda$'s evolution. As $\lambda$ adapts to specialize on a novel receptor, the fitness landscape shifts to allow for the emergence and coexistence of a previously inaccessible phenotype.

We observed this phenomenon of adaptive dynamics in a prokaryotic virus diversifying host receptor tropism between two *E. coli* outer membrane proteins, but this process likely occurs in other viruses adapting to new hosts. Diversification of receptor tropism can lead to cross-species transmission and pandemics[33], and appropriate tropism is necessary for therapeutic application of viruses to treat human disease in the emerging fields of phage therapy[34], gene therapy[35], and oncolytic virotherapy[36]. A better understanding of how a given virus's fitness landscape is reshaped by epistasis, pleiotropy, and competition for resources across varying environments may more generally provide an additional framework for understanding the natural adaptation of pathogenic viruses to new niches, and may help unlock the adaptive potential of therapeutic viruses through directed evolution. However, we do not suspect that these principles of adaptive evolution are unique to the virus world. It is tempting to speculate that the effects that dynamic fitness landscapes exert on biological diversification may be generalizable across all domains of life.

## Methods

### Cells, viruses, and media

Bacterial hosts used for evolution and competition experiments were the *LamB* ("L⁻" strain JW3996) and *OmpF* ("O⁻" strain JW0912) knockout strains from the Keio collection[37]. The "wild type" *E. coli* K-12 BW25113 (parental strain in the Keio collection) was used unless otherwise stated to titer $\lambda$ phages by plaque assay using standard protocols. The host strain HWEC106, which contains a mutation in the *mutS* mismatch repair gene and the pKD46 plasmid containing the arabinose-inducible $\lambda$ red recombineering machinery[38], was used to perform mutagenesis and is the strain that we used to maintain the cI26-derived engineered lysogen library as described below. Speciation replay experiments used strictly lytic phages derived from the $\lambda$ phage strain cI26. The receptor generalist which is used as the ancestor for this experiment (termed "EvoC") contains five mutations in *J* relative to cI26[19] which permit the use of the second receptor OmpF in addition to the native receptor LamB. The virus strain used to create the mutant virus library used for competition experiments is described in the methods for "Lysogen library mutagenesis" below. Standard LB Lennox media was used to culture *E. coli* in the absence of phage infection; a modified media "LBM9" was used when inducing or infecting with phage (20 g tryptone, 10 g yeast extract per liter of water, supplemented with 47.7 mM disodium phosphate heptahydrate, 22 mM potassium phosphate monobasic, 18.7 mM ammonium chloride, 8.6 mM sodium chloride, 0.2 mM calcium chloride; immediately prior to use, the media was supplemented with 10 mM magnesium sulfate).

### Speciation replay experiment with population sequencing

We replicated our previous speciation experiment[18] but with slight changes. Briefly, the ancestor receptor generalist strain (EvoC) evolved by daily serial passage of the phage population into an even mixture of fresh L⁻ and O⁻ host cells. Each day, after 8 hours of culture, phage was isolated from bacteria by centrifugation and filter sterilization of a 10 ml culture onto tubes kept on ice with 50 $\mu$l chloroform (as a preservative). The following day, fresh overnight cultures of L⁻ and O⁻ cells were diluted 1:10 and allowed to grow for two hours to reach the exponential growth phase, then further diluted 1:5000 and added with 100 $\mu$l of isolated phage to a final volume of 10 ml with LBM9 and cultured again for 8 hours in a 50-ml Erlenmeyer flask for at 37°C with shaking at 120 rpm before again isolating phage, and the process was repeated daily. Every five days, phage genomes were extracted in bulk and *J* amplicon sequencing was performed with 150-bp paired end reads using an Illumina iSeq 100 and mutation frequencies were computed using breseq[39] set to polymorphism mode with default settings; similar results were obtained with less strict filtering parameters. Specialist phages (L-specialist and O-specialist) were isolated by plaque purification (on O⁻ cells and L⁻ cells, respectively), and the specialization index was measured for these two isolated specialist phages using standard plaque-formation assays on L⁻ and O⁻ cells, by first infusing the appropriate host strain ( ~ 5×10⁹ cells) into 10 ml of molten soft agar (10 g tryptone, 1 g yeast extract, 8 g sodium chloride, 7 g agar, 1 g glucose, 1.23 g magnesium sulfate per L water) at 55°C and then plating the infused soft agar over a 15-cm diameter Petri dish preplated with a standard LB agar base; after the soft agar had solidified (approximately 5–10 minutes), 2 $\mu$l drops of 10-fold serial dilutions were spotted onto the surface[18].

### Lysogen library mutagenesis

Our prior work studying receptor specialization in $\lambda$ was done using the strictly lytic strain cI26[18], which has a nonsense mutation in the lysis repressor gene *cI* and lacks the functional *attp* integration site. For the purposes of genetic engineering of the $\lambda$ genome, we engineered a lysogenic form of cI26 by replacing the nonfunctional *cI* gene with a temperature-sensitive mutant (to enable heat induction of the lytic pathway) and adding a constitutively expressed chloramphenicol

resistance gene for antibiotic selection of lysogens in the bacterial host. Stable lysogens were maintained at 30°C in host strain HWEC106 in the presence of chloramphenicol, and infectious phage was induced at 37°C using standard heat induction protocols.

Prior to introducing the combinatorial library mutations, in-frame stop codons were introduced in the *J* coding sequence at the 5' (overwriting nucleotides 2989-2995 with 'TGATAGT') and 3' (overwriting nucleotide 3232 with 'T') edges of the region undergoing mutagenesis, so that lysogens that did not undergo recombination with mutagenic oligonucleotides would not produce functional viruses. This step was necessary to avoid the mutant virus library being comprised of primarily unmutated genotypes, since only a small proportion of lysogens undergo recombination. Mutations were introduced using MAGE[17,40]. Briefly, we performed iterative rounds of mutagenesis by electroporating a mixture of long double-stranded DNA fragments encoding the combinatorial library into cells with an induced $\lambda$ red recombination system. The fragments were generated from PCR amplification of 300nt-long single-stranded DNA oligo pool from Twist Biosciences. Lysogen libraries were created in duplicate (Library replicates A and B). Because the ninth mutation site was too far away from the other eight to create a combinatorial library for all nine mutation sites in one shot, two sub-libraries (#1 and #2) were created for each replicate (A and B); sub-libraries 1 A and 1B were constructed without the ninth mutation, and sub-libraries 2 A and 2B were constructed on a starting sequence containing the ninth mutation. Each sub-library contains $2^8 = 256$ genotypes, and the combination of the two libraries contains the full combinatorial sampling across all 9 sites of $2^9 = 512$ genotypes. Deep sequencing of the lysogen sub-libraries confirmed that not only were all 256 expected genotypes introduced in all replicates (Supplementary Table 1), but that they were introduced in relatively uniform frequencies (Supplementary Fig. 2a).

Competitor virus genotypes were constructed by $\lambda$ red recombineering, by introducing the *J* mutations of the L-specialist or the O-specialist from the speciation replay experiment (Supplementary Fig. 1), or the ancestral EvoC sequence, into the same engineered lysogenic cI26 strain as above. The three competitor virus genotypes also encoded synonymous mutations (c2979a, c2982a, g2985t, t2991c, a2994c, and g2997a) at the forward primer binding site used in the 'round 1 PCR' step below (the step attaching partial Illumina sequencing adaptors), thus excluding PCR amplification of the competitor viruses during sequencing library preparation such that only the library viruses, and not the competitor viruses, were captured in deep sequencing.

## Lysogen library induction and amplification of virus libraries

Because the lysogen libraries (by nature of their construction described above) are predominantly encoding nonviable *J* genes containing stop codons flanking the mutagenesis region, only a small fraction of lysogen cells produce viable mutant viruses upon heat induction. Care was taken during induction and amplification by using larger volumes containing larger numbers of viral particles at each step, to avoid bottlenecking library complexity. Lysogen sub-libraries A1, A2, B1, and B2 were grown overnight in LB supplemented with 25 $\mu$g/ml chloramphenicol at 30°C. To induce lysogens growing exponentially, 133 $\mu$l of overnight lysogen culture was diluted into 4 ml of LBM9 into 10-ml culture tubes and grown for two hours with shaking at 220 rpm at 30°C. These exponentially growing cultures were subjected to heat shock at 42°C for 15 minutes followed by incubation with shaking at 220 rpm at 37°C for 90 minutes to allow for lysis and release of $\lambda$ virions. The lysates from 10 replicate induction tubes for each sub-library (40 ml total lysate per sub-library for sub-libraries 1 A, 1B, 2 A, and 2B, all induced separately) were combined and filtered through 0.22 $\mu$ m syringe filters. These lysates were at a relatively low titer of infectious virions (~5 ×10³ PFU/ml lysate by plaquing assay on *E. coli* K

-12) so we then amplified each sub-library for four hours on *E. coli* K-12 cells at 37°C (in four replicate flasks per sub-library to maintain library diversity) by adding 8.5 ml of lysate with 0.5 ml of overnight K-12 culture and 1 ml of LBM9, before finally filtering the lysates through a 0.22 $\mu$ m syringe filter, combining the lysates from each four replicate flasks of amplification for each sub-library, and freezing 1 ml aliquots of amplified sub-libraries A1, A2, B1, and B2. The amplified sub-libraries reached an average titer of ~5 ×10⁸ PFU/ml. The final virus libraries "A" and "B" were then made by mixing equal plaque-forming units of amplified sub-libraries A1 + A2 and B1 + B2, respectively.

## Competition experiments to measure fitness landscapes

$1 \times 10^6$ PFU of the specified virus library was mixed with 9 ×10⁶ PFU of the specified competitor virus (ancestor EvoC, L-specialist, or O-specialist) and incubated with 2 ×10⁸ CFU of the specified host cells (either L⁻ or O⁻ cells, or an equal mixture of the two), resulting in a multiplicity of infection of 0.05, in a total 10-ml culture in 50-ml flasks using LBM9 media. Infection proceeded for four hours shaking at 3 °C (roughly two $\lambda$ infection cycles) prior to filtering the lysates and extracting viral genomes for deep sequencing. Each independent library replicate "A" and "B" was tested in triplicate competition flasks, each of which was independently sequenced and analyzed as described below. Therefore, in total, for each of the five competition environments (competed against generalist on L⁻, O⁻, or equal mixture of hosts, competed against L-specialist on equal mixture of hosts, and competed against O-specialist on equal mixture of hosts), there were six replicate competitions (three for each replicate library).

## Phage genomic DNA extraction

Filtered lysates (~10 ml) from each competition experiment were mixed with 6.5 ml of phage precipitation buffer (20% w/v PEG 6000, 2.5 M NaCl, Teknova P4168) and incubated overnight at 4 °C. Phage were pelleted at 10,000x*g* for 30 minutes at 4°C and the virus pellets were gently resuspended in 360 $\mu$l phage resuspension buffer (1 M NaCl, 10 mM Tris pH 7.5, 0.1 mM EDTA) and transferred to a 1.7 ml microcentrifuge tube. 55 $\mu$l DNase I 10X buffer, 1 $\mu$l DNase I (2000 U/ml), and 1 $\mu$l RNase (20 mg/ml) were added to samples and incubated at 37°C for 30 minutes. 10 $\mu$l 0.5 M EDTA was added to stop DNase activity prior to adding 2.5 $\mu$l proteinase K (20 mg/ml) and 25 $\mu$l 10% SDS and incubating at 55°C for 60 minutes to break apart virions. Phage genomic DNA was extracted with 600 $\mu$l Tris-saturated phenol-chloroform-isoamyl alcohol, followed by two chloroform extractions to remove residual phenol from the aqueous phase. Finally, 50 $\mu$l 3 M sodium acetate and 1000 $\mu$l pre-chilled 100% ethanol were added and samples were incubated at -20°C overnight for DNA precipitation. DNA was pelleted by centrifugation at 14,000x *g* for 30 minutes at 4°C, gently washed with 500 $\mu$l of pre-chilled 70% ethanol, pelleted again at 14,000x *g* for 5 minutes, pellets gently air-dried, and resuspended in 50 $\mu$l DNA elution buffer (10 mM Tris-Cl, pH 8.5) and concentration quantified using Qubit 1X dsDNA HS Assay Kit.

## Deep sequencing of mutant virus libraries

PCR was used to generate Illumina sequencing libraries from each phage genomic DNA extraction. All PCR mixtures used Q5 Hot Start High-Fidelity 2X Master Mix (NEB) with the specified primers and templates below. PCR products were purified using CleanNGS DNA & RNA Clean-Up Magnetic Beads (Bulldog Bio) and quantified using Quant-iT PicoGreen (Thermo Fisher). First a PCR amplicon of the *J* gene containing the 9 sites of mutations in the library was generated using primers JRR-F (5'- GGAAAGCTGACCGCTAAAAATGC -3') and JRR-R (5'- TAAAACGCCCGTTCCCGGAC -3'), each at final concentration 0.5 μM, with 200 ng phage genomic DNA template, in a final volume of 50 $\mu$l, using cycling parameters: 98 °C for 2 minutes, then 25 cycles of 98 °C for 10 seconds, 68 °C for 15 seconds, 72 °C for 10 seconds; followed by a final extension step at 72 °C for 2 minutes.

Next, a two-round PCR procedure was used to add unique molecular identifier barcodes and Illumina adapter sequences. Round 1 PCR used the following primers to add unique molecular barcodes and partial Illumina adaptor sequences to J amplicon: R1-Reverse (5′-GGAGTTCAGACGTGTGCTCTTCCGATCT-X-CGGCGGAATTTTTGCCG-3′) and R1-Forward (5′- CTTTCCCTACACGACGCTCTTCCGATCT -X -TCGTCGGGGAAATTGTAAAG -3′), where 'X' represents a string of 5-12 random (N) bases, with lengths of 5-12 mixed in equimolar ratios for both forward and reverse primers. For lysogen sub-library sequencing, a different R1-Reverse primer was used (5′- GGAGTTCAGACGTGTGCT CTTCCGATCT-X-CTGGCATGTCAACAATACGG-3′) resulting in a shorter amplicon which did not contain the ninth mutation site, but was able to be sequenced with shorter (100 bp) sequencing reads. Of note, during this PCR step, the competitor viruses were excluded by amplification by synonymous mutations within these primer binding sites. Lysogen sub-libraries containing the ninth mutation (2 A, 2B) and without the ninth mutation (1 A, 1B) were independently sequenced to confirm allelic representation across the other eight mutation sites after mutagenesis (Supplementary Table 1, Supplementary Fig. 2a).

Round 1 PCR was performed on 4 ng of purified J amplicon template, each primer at 0.4 μM, in final volume of 25 μl, using the following thermocycling parameters: 2 minutes denaturation at 98 °C; 11 cycles of amplification with 98 °C 10 sec, 63°C 15 sec, 72°C 10 sec; with final extension step at 72 °C for 2 minutes, followed by denaturation at 95 °C for 1 minute. The final denaturation step ensures that each double-stranded DNA product molecule contains two single strands with unique molecular barcodes. Round 1 PCR products were purified, quantified, and diluted to control the number of unique barcodes used in round 2 PCR to ensure that expected read depth would capture each barcode multiple times to build error-corrected consensus sequences.

Round 2 PCR used approximately $2 \times 10^6$ double-stranded DNA molecules of purified round 1 products as template (i.e., $4.4 \times 10^{-4}$ ng of 433 bp dsDNA), round 2 primers each at 0.5 μM (R2-Forward: 5′- AAT GATACGGCGACCACCGAGATCTACACTCTTTCCCTACACGACGCTCTT CC -3′; R2-Reverse: 5′- CAAGCAGAAGACGGCATACGAGAT-X-GTGACT GGAGTTCAGACGTGTGCTCTTCCGATCT-3′; where 'X' is replaced with sample-specific index sequences for multiplexing), in a final volume of 40 μl, using the following thermocycler parameters: 98°C for 2 min, 25 cycles of (98 °C 10 sec, 71 °C 15 sec, 72 °C 10 sec), and final extension step at 72 °C for 2 minutes. Competition experiments were sequenced with 150 bp paired-end reads on a NovaSeq 6000 at the UCSD IGM Genomics Center. Lysogen sub-library genomic DNA samples were prepared and sequenced separately (as sub-libraries) using a shorter amplicon length with 100 bp paired-end reads.

## Parsing genotype counts from FASTQ files and calculation of selection rate

Custom python scripts were used to parse FASTQ files using pysam 0.21.0 (https://github.com/pysam-developers/pysam), aligning reads to the J gene, parsing the unique molecular barcode, building consensus sequences for each molecular barcode observed by at least 3 independent reads, and tabulating the number of unique molecular barcodes observed for each J genotype sequence. The observed frequencies for all genotypes at time t = 0 and time t = 4 hours were then used to calculate the selection rate S, per unit time of 4-hour competition experiment[17], providing a fitness metric relative to the ancestor generalist sequence EvoC:

$$S = \frac{\ln(G_T/G_0) - \ln(EvoC_T/EvoC_0)}{T} \quad (1)$$

where $G_T$ and $EvoC_T$ are the frequencies of a given genotype G and EvoC at time point T, respectively. In this framework, since the ancestor EvoC sequence is the reference for defining selection rates, the selection rate of EvoC is always zero.

We discovered that an additional mutation (c3283t) was present at modest amounts and in association with many of the combinations of the programmed mutations introduced by MAGE (Supplementary Fig. 2b–c, teal bars). We infer (but are not certain) that this mutation was present at modest frequencies in the lysogen stocks prior to mutagenesis, because we also observed it in linkage with the stop codons that are present in the parental strain and overwritten during mutagenesis. We analyzed the fitness of genotypes that were observed with and without this mutation and found them to be modestly correlated in all environments (Supplementary Fig. 8). The remainder of the analysis was limited to genotypes that only contained the programmed 512 combinations of mutations in the library design. The distribution of fitness effects for each of the five fitness landscapes is provided in Supplementary Fig. 14.

## Linear regression

To detect which genetic and environmental factors shape the fitness landscape, we used a linear regression model where each genotype's fitness was predicted as a function of its J mutations and the identity of the competitor virus present (generalist, L-specialist, or O-specialist). The predictor variables included terms for single mutations ('G' terms), additional non-additive epistasis amongst pairs of mutations using mutation-by-mutation terms ('GxG' terms), competitor terms independent of genotype ('C' terms), competitor-dependent terms for single and paired mutations ('CxG' and 'CxGxG' terms), and an intercept. Because mutations at sites 2 and 3 introduce the same amino-acid change (either alone or in combination) and because the measured fitness values were highly correlated between genotypes containing these synonymous variations (Supplementary Fig. 6 and 7), we used a single term in the regression model to indicate the presence of either mutation 2 or mutation 3 or the combination of the two. In total, there were 148 features in the model: 1 intercept, 8 G, 3 C, 28 GxG, 24 CxG, and 84 CxGxG terms.

We only used fitness measurements from the competition experiments performed on a mixture of L⁻ and O⁻ host cells, to limit the model to predicting fitness as a function of genotype and competitor alone. In total we performed 6 replicate competition experiments (three with each independent library) in each of the three environments (against an L-specialist, against an O-specialist, and against a generalist), for 18 total genotype fitness datasets. Each dataset contained fitness observations for between 345 and 467 genotypes (out of 512 possible; some genotypes are not observed in some samples), and thus a total of 7375 fitness measurements were used in the regression model. Linear regression was performed using the scikit-learn python module (version 1.3.0), using the LassoLarsIC function to perform LASSO regularization to reduce model complexity. The regularization parameter was tuned using multiple approaches: AIC (Akaike information criterion), BIC (Bayesian information criterion), and cross-validation. Similar results were obtained for each approach.

## Computing receptor specialization for all genotypes in the library

Selection rates computed from library competition experiments performed in the presence of only O⁻ or L⁻ host cells were used as measurements for each genotype's fitness on each of the individual receptors LamB and OmpF, respectively. Specialization index (SI) was then computed for each genotype using the selection rate on each receptor using an equation analogous to our prior work[18] but using selection rates, which when exponentiated, are proportional to plaque-forming units:

$$SI = \frac{e^{S_L} - e^{S_O}}{e^{S_L} + e^{S_O}} \quad (2)$$

where $S_L$ and $S_O$ are the selection rates on LamB and OmpF, respectively. The specialization index ranges from -1 (complete O-specialist)

to +1 (complete L-specialist). Zero represents equal fitness on both receptors. The selection rate for the ancestral strain used in these experiments, EvoC, is zero by default since it is the reference strain against which all the other selection rates are measured against, and thus the specialization index of the EvoC genotype is, by definition, also zero.

## Analysis of hybrid incompatibility

We limited our analysis of the effects of hybridization to L-specialists and O-specialists that all had selection rates measured on both receptors (e.g., on L⁻ cells alone and on O⁻ cells alone) so that we could implement a geometric interpretation of the effects of hybridization on fitness (Supplementary Fig. 4a). We used a conservative threshold in SI to categorize some genotypes as being receptor specialists for this analysis, since there is no conventional quantitative definition of specialists in this context. We defined receptor specialists as those genotypes with absolute SI > 0.33. We chose this cut-off value because based on the formula above for SI, a cut-off value of SI = 0.33 defines a specialist genotype as one that has a growth rate approximately two (or more) times greater on the specialized receptor than the non-specialized receptor. Using this definition, we identified 27 genotypes classified as L-specialists that had observed selection rates on both O⁻ and L⁻ cells. We analyzed these L-specialist genotypes along with the top 27 most specialized O-specialist genotypes (Supplementary Fig. 4b). We iterated over all 729 pairwise combinations of an L-specialist and O-specialist from these groupings and computed a hybrid genotype that combined the mutations from each of the chosen specialists. In some cases, the hybrid genotype is actually the same as one of the specialist genotypes, and in other cases the hybrid genotype did not have an observed selection rate on one of the host cell types; after subtracting these from the analysis, 536 specialist-specialist-hybrid trios remained. For all trios we computed the magnitude of a vector that is orthogonal to, and arising from, the line connecting the two specialist genotypes on the axis depicted in Supplementary Fig. 4a to the hybrid genotype. The magnitude of this vector measures the change in average receptor fitness in the hybrid genotype relative to the two parental genotypes. Positive and negative values denote improved or worsened average receptor fitness in the hybrid relative to the two parents.

## Computer simulations of λ evolution

We simulated λ evolution using a modified Wright-Fisher model similar to our previous study[17]. This model incorporates the effects of mutation, selection, and genetic drift to model a virus population as it samples mutations at the nine sites in J. Briefly, the model uses a fixed population size (in our simulations, $5 \times 10^9$ viral particles) and a pure starting genotype (the ancestor receptor generalist, EvoC). The number of progeny viral particles in each new generation are drawn from a multinomial distribution, where the probability for each genotype to reproduce is proportional to the product of the abundance of viral particles of that genotype and the measured fitness of that genotype. Random mutations introduced at each generation, based on the estimated mutation rate of λ ($7.7 \times 10^{-8}$ substitutions per base per replication[41]), provided the chance for new genotypes to emerge. Representative population curves under various models are shown in Supplementary Fig. 5.

We ran each simulation for 500 generations. The generation time of λ in our experiments is difficult to know precisely. Generation time depends on adsorption rate (which itself depends on J genotype[14] and host and phage densities during the course of the experiment) and on lysis time, which exhibits some degree of variability[42]. Using an approximation of 60 minutes as the generation time, each eight-hour 'day' in the speciation replay experiment would reflect 8 generations, resulting in a total of $8 \times 35 = 280$ generations in the replay experiment. We conservatively chose to run our simulations for 500 generations to

ensure that the simulations would be long enough to sufficiently capture evolutionary dynamics occuring in our replay experiment. In most cases across all models, a state of equilibrium is reached by generation 280 and endures until generation 500 (Fig. 4a, Supplementary Fig. 5, Supplementary Fig. 12).

Since our measurements of fitness exhibited some degree of uncertainty between experimental replicates, we designed the simulations such that replicate simulations incorporated a similar degree of uncertainty by adding noise to fitness landscapes on a per-simulation basis. Each simulated evolution experiment used fitness landscapes drawn from a distribution of landscapes centered on the average of the experimental measurements but with normally distributed noise applied to each fitness value. The magnitude of the noise was tuned to produce replicate draws of noise-added fitness landscapes with correlation coefficients similar to the correlations between experimental replicates (Pearson correlation coefficient for selection rates measured between two independent virus libraries ranged from 0.95 to 0.98, Supplementary Fig. 3, Supplementary Fig. 9).

We ran simulations under seven different models that differed in how the various fitness landscapes were used, and for each model we ran 500 replicate simulations. Five models each used a static fitness landscape that remained constant across all generations (corresponding to the five different environments in which the landscapes were measured, see Fig. 3), and two models used a shifting fitness landscape to account for changes in the fitness landscape that are induced by the emergence of new competing phenotypes in the viral population.

For the shifting model simulations, we initially ran a 'continuous' shiting model as follows: the five experimentally measured landscapes were positioned along a 'landscape axis' (Supplementary Fig. 10) at coordinates defined by the population SI (average of SI across all genotypes in the population, weighted by abundance) associated with the experimental conditions used to measure the landscapes. We computed the population SI for the experimental conditions used to measure the fitness landscapes as follows: in the three landscapes measured with both L⁻ and O⁻ host cells (Fig. 3b–d), population SI was computed with the assumption that the mutant virus library (which by design was present as 10% of the virus population) has a net SI of 0 (as it is a complex mixture of genotypes sampling both generalists and specialists to varying degrees), and the remainder 90% of the population (by experimental design) was comprised of a competitor genotype with a known SI as computed in Fig. 2b (SI for the L-specialist competitor = +0.778, for the generalist EvoC = 0, and for the O-specialist competitor = -0.99). By calculating the population SI for each of these experimental conditions, we positioned landscape B at population SI = +0.7, landscape C at population SI = 0, and landscape D at population SI = -0.9; as these were the population SI present in the experimental conditions that were used to measure each of these landscapes. For the two extreme ends of the landscape axis, we positioned the landscape measured with only OmpF-expressing hosts (L⁻ cells) at +1, and the landscape measured with only LamB-expressing hosts (O⁻ cells) at -1, since these two experimental conditions approximate the limiting cases of saturation with an L-specialist (or O-specialist) to the extreme degree that LamB-expressing hosts (or OmpF-expressing hosts, respectively) are no longer available for infection. While simulating evolution on the continuous model, at each generation the population SI was calculated and the fitness landscape used to govern reproduction probabilities in that generation was interpolated as a weighted average of the two experimental landscapes nearest to the population SI on the landscape axis.

To test whether the results of the continuous shifting model were dependent on the method of continuous interpolation between experimentally measured landscapes, we also implemented a 'discrete' shifting model. The discrete model only used the five experimentally

measured landscapes and shifted between landscapes based on which landscape was 'closest' on the landscape axis to that generation's population SI. Only the continuous model is shown in Fig. 4 for brevity; the effect on maintaining genetic and phenotypic diversity is the same in both shifting models (Supplementary Fig. 11).

For all evolutionary models, the Shannon entropy was computed for each generation from the abundances of all genotypes in that generation. Phenotypic diversity at the endpoint of each simulation was summarized by counting the number of discrete phenotypes (L-specialist, O-specialist, or generalist) present at the endpoint across all genotypes present with at least 2.5% abundance. Trajectories of population SI for all 500 simulations within each of the total seven evolutionary models are provided in Supplementary Fig. 12, and the corresponding median and interquartile range of SI trajectory for each model is shown in Supplementary Fig. 13.

### Reporting summary
Further information on research design is available in the Nature Portfolio Reporting Summary linked to this article.

## Data availability
Sequencing data are available from the Sequence Read Archive under BioProject PRJNA984687, BioSample Accession SAMN35774826. All data analyses and intermediate datasets are deposited in a GitHub repository (github.com/mbdoud/shifting-fitness-landscapes) and archived at Zenodo DOI: 10.5281/zenodo.10459906 (https://doi.org/10.5281/zenodo.10459906)[43].

## Code availability
The computer code for computational analyses and figure generation is available in a GitHub repository (github.com/mbdoud/shifting-fitness-landscapes) and has been deposited at Zenodo https://doi.org/10.5281/zenodo.10459906 (https://doi.org/10.5281/zenodo.10459906)[43]. The computational analyses were run using python 3.11.4, numpy 1.25.1, scipy 1.11.1, scikit-learn 1.3.0, pandas 2.0.3, networkx 3.1, matplotlib 3.7.2, and pysam 0.21.0.

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

## Acknowledgements

This work was supported by NSF grant 1934515 to JM and NIH grant T32AI007036 to MD. This publication includes data generated at the UC San Diego IGM Genomics Center utilizing an Illumina NovaSeq 6000 that was purchased with funding from a National Institutes of Health SIG grant #S10 OD026929.

## Author contributions

M.B.D., A.G., and J.R.M. conceptualized the study. M.B.D., A.G., V.L., S.J.M., and C.D.L.F performed experiments. V.L. analyzed sequencing data from the speciation replay experiment. M.D. conducted all other data analyses and made figures. J.R.M. supervised the study. M.B.D. and J.R.M. wrote the manuscript, and all authors edited the manuscript.

## Competing interests

The authors declare no competing interests.
