## [Peer Review File · Nature Communications]

Competition-driven eco-evolutionary feedback reshapes bacteriophage lambda's fitness landscape and enables speciationReviewers' Comments:

Reviewer #1:

Remarks to the Author:

In this study the authors study how competition between two virus specialists shapes the fitness landscape. To start they repeat a speciation experiment in lambda virus that is adapting to different host cells. Using mutations identified during evolution to the different host cells they build the fitness landscape for these 9 mutations and test their effect in different environments: presence of different competitors and different host cells. Finally, they simulate evolution using the empirical genotype-phenotype-map with static and dynamic fitness landscapes, that depend on the specialization index of the virus. They show that both the fitness effect of single mutations, as well as their epistatic effect change in response to the presence of competitor and host cell frequency. In addition, they show that considering simulated evolution in dynamic fitness landscapes (instead of static landscapes) allows to better to describe the dynamics observed in the real experiment. This is, to my knowledge, one of the first studies that empirically shows that competition changes fitness landscape dynamically.

This is a very interesting study using a really amazing system to study dynamic fitness landscapes. The results and methods are easy to read and provide a very detailed explanation of the different experiments and combination of simulations and empirical data. Overall, the manuscript is well written.

However, there is no discussion of the literature in the study, the results and discussion section although providing a very thorough overview of the results, do not put them into context of what is known of the literature and how the results compare and advance our knowledge. This is also seen in the introduction. Thus, most of my comments are about what literature is missing in the different sections to answer the questions posed by the authors. I focused on the discussion, because it is the easiest to comment, but then the introduction should be updated accordingly. Comments are provided in order of appearance.

Introduction - The introduction needs a restructure. The idea is super nice, but it is only understood in the final part of the introduction. Maybe it would make the intro flow better to start by explaining fitness landscapes and what is thought to happen during adaptation/speciation. Then introduce the adaptive dynamics and talk about previous work? There is very little information about all the work that has been done in viruses and fitness landscape. Finally, although the study system is important, this should be done in a smaller paragraph together with the specific questions of the study.

L30: 44 - The first two sections of the introduction seem to be trying to strongly motivate why virus are interesting and can be used to study this topic. However, typically the introduction starts with the general picture of the main topic of the study (e.g. competition driving evolution or fitness landscapes) and its relevance should be presented here. The motivation to use virus can be introduced later.

L68: what key prediction?

L76:79 – This sentence is complex and should be rephrased. Maybe breaking it in two, would make it clearer?

L81 - remove phenotype from the sentence.

L100: 101 - "We repeated the experiment to ensure repeatability in a new location and to preserve samples every 40 hours to track evolutionary dynamics." - The end of this sentence does not make sense, maybe there is a verb missing?

L101: J gene

L107: "All nine mutations resulted in nonsynonymous changes" - Later on, mutations 2 and 3 are

coupled together because they introduce the same amino acid change (together or alone). So at least these 2 mutations are synonymous. Is this correct?

L110 - This title is uninformative (in comparison with the other titles), maybe a small title summarizing the results from this section would be better, for example: High repeatability of the fitness landscape that leads to virus speciation?

L 145 - The selective coefficient was calculated in one unit of time, but this corresponds to around two generations. Why not use the expected number of generations of the wild type instead of assuming one time unit? Alternatively, using the number of hours. This allows to scale better the effect of the mutant in relation to the wild type.

See also:

Matuszewski, S., Hildebrandt, M. E., Ghenu, A. H., Jensen, J. D., & Bank, C. (2016). A statistical guide to the design of deep mutational scanning experiments. *Genetics*, 204(1), 77-87.

L154: 174 - This section is only results, but there are already several papers that tackle how epistasis is environment dependent and can be change by presence of competitors.

Some references that may be interesting to look at for this:

- Hall, A. E., Karkare, K., Cooper, V. S., Bank, C., Cooper, T. F., & Moore, F. B. G. (2019). Environment changes epistasis to alter trade-offs along alternative evolutionary paths. *Evolution*, 73(10), 2094-2105.- Baier et al 2023 Science
- Lalić, J., & Elena, S. F. (2013). Epistasis between mutations is host-dependent for an RNA virus. *Biology letters*, 9(1), 20120396.
- Flynn, K. M., Cooper, T. F., Moore, F. B., & Cooper, V. S. (2013). The environment affects epistatic interactions to alter the topology of an empirical fitness landscape. *PLoS genetics*, 9(4), e1003426.
- Bank, C. (2022). Epistasis and adaptation on fitness landscapes. *Annual review of ecology, evolution, and systematics*, 53, 457-479.
- Cervera, H., Lalić, J., & Elena, S. F. (2016). Effect of host species on topography of the fitness landscape for a plant RNA virus. *Journal of Virology*, 90(22), 10160-10169.

L155- The initial sentence of the paragraph should introduce the reader to why is it important to study the effect of epistasis. Currently, it reads more as a title of a sub-section than an initial sentence.

L175: 186 – In this section it would be important to define specialists and generalists. There is already a plethora of literature that talks about this. A good place to start would be to see:

- Elena SF, Agudelo-Romero P, Lalić J. The evolution of viruses in multi-host fitness landscapes. *Open Virol J*. 2009 Mar 19;3:1-6. doi: 10.2174/1874357900903010001. PMID: 19572052; PMCID: PMC2703199.
- Kassen, R. (2002). The experimental evolution of specialists, generalists, and the maintenance of diversity. *Journal of evolutionary biology*, 15(2), 173-190.
- Syller, J., & Grupa, A. (2016). Antagonistic within-host interactions between plant viruses: molecular basis and impact on viral and host fitness. *Molecular plant pathology*, 17(5), 769-782.

L203: 206 - Here makes sense to discuss DMI in respect to what is known in the literature. Are they prevalent overall? is this common in virus? Some references that may be useful:

- Duffy S, Burch CL, Turner PE. Evolution of host specificity drives reproductive isolation among RNA viruses. *Evolution*. 2007 Nov;61(11):2614-22. doi: 10.1111/j.1558-5646.2007.00226.x. Epub 2007 Oct 1. PMID: 17908251; PMCID: PMC7202233.
- Zhao L, Seth-Pasricha M, Stemate D, Crespo-Bellido A, Gagnon J, Draghi J, Duffy S. Existing Host Range Mutations Constrain Further Emergence of RNA Viruses. *J Virol*. 2019 Feb 5;93(4):e01385-18. doi: 10.1128/JVI.01385-18. PMID: 30463962; PMCID: PMC6364021.
- Paixão, T., Bassler, K. E., & Azevedo, R. B. (2014). Emergent speciation by multiple Dobzhansky–Muller incompatibilities. *bioRxiv*, 008268.
- Unckless RL, Orr HA. Dobzhansky-Muller incompatibilities and adaptation to a shared environment.

Heredity (Edinb). 2009 Mar;102(3):214-7. doi: 10.1038/hdy.2008.129. Epub 2009 Jan 14. PMID: 19142201; PMCID: PMC2656211.

L223: 225 - How does this compare what is described in the literature? is this common? There aren't many studies, especially in viruses, but maybe looking to other systems would also be important.

Some references:

- Martin, C. H., & Wainwright, P. C. (2013). Multiple fitness peaks on the adaptive landscape drive adaptive radiation in the wild. *Science*, 339(6116), 208-211.
- Rainey, P. B., & Travisano, M. (1998). Adaptive radiation in a heterogeneous environment. *Nature*, 394(6688), 69-72.
- Hendry, A. P., Nosil, P., & Rieseberg, L. H. (2007). The speed of ecological speciation. *Functional ecology*, 21(3), 455.

L226: 292 – This final section, should wrap up the different predictions done by dynamic/shifting landscapes and static landscapes. What is known about fitness landscapes when other species evolve. The few work that has been developed are mostly about host-parasite coevolution, but still it would be important to discuss what are the differences as similarities and put them into context.

Some references that can be useful:

- Amado, A., & Bank, C. (2023). Ecological tradeoffs lead to complex evolutionary trajectories and sustained diversity on dynamic fitness landscapes. *bioRxiv*, 2023-10.
- Peri, G., Gibard, C., Shults, N. H., Crossin, K., & Hayden, E. J. (2022). Dynamic RNA fitness landscapes of a group I ribozyme during changes to the experimental environment. *Molecular biology and evolution*, 39(3), msab373.
- Rubin, I. N., Ispolatov, Y., & Doebeli, M. (2023). Adaptive diversification and niche packing on rugged fitness landscapes. *Journal of Theoretical Biology*, 562, 111421.
- Bajić, D., Vila, J. C., Blount, Z. D., & Sánchez, A. (2018). On the deformability of an empirical fitness landscape by microbial evolution. *Proceedings of the National Academy of Sciences*, 115(44), 11286-11291.
- Martin, C.H. and Gould, K.J. (2020), Surprising spatiotemporal stability of a multi-peak fitness landscape revealed by independent field experiments measuring hybrid fitness. *Evolution Letters*, 4: 530-544. <https://doi.org/10.1002/evl3.195>
- Patton, A. H., Richards, E. J., Gould, K. J., Buie, L. K., & Martin, C. H. (2022). Hybridization alters the shape of the genotypic fitness landscape, increasing access to novel fitness peaks during adaptive radiation. *Elife*, 11, e72905.
- Gavrillets, Sergey, 'High-Dimensional Fitness Landscapes and Speciation', in Massimo Pigliucci, and Gerd B. Müller (eds), *Evolution—the Extended Synthesis* (Cambridge, MA, 2010; online edn, MIT Press Scholarship Online, 22 Aug. 2013), <https://doi.org/10.7551/mitpress/9780262513678.003.0003>,
- Braga, M. P., Araujo, S. B., Agosta, S., Brooks, D., Hoberg, E., Nylin, S., ... & Boeger, W. A. (2018). Host use dynamics in a heterogeneous fitness landscape generates oscillations in host range and diversification. *Evolution*, 72(9), 1773-1783.
- Williams, H.T. Phage-induced diversification improves host evolvability. *BMC Evol Biol* 13, 17 (2013). <https://doi.org/10.1186/1471-2148-13-17>
- Aguirre, J., Catalán, P., Cuesta, J. A., & Manrubia, S. (2018). On the networked architecture of genotype spaces and its critical effects on molecular evolution. *Open biology*, 8(7), 180069.

L231: This seems like a weird sentence. Either be specific of how many mutations, or remove the only.

L249: 250 – When I read this section, my main comment was that it is not clear how is it possible to decouple the observations from the expectation? i.e. if the fitness landscape is computed by the weighted average of the population, then of course that it will show that the population is shifting. The genotype-phenotype-fitness map should be fixed.

However, when reading the methods this became clearer, so to be sure, you computed the FL from the experimentally obtained data, but assuming a prevalence of competitor of 90% and 10% of a mix and then in the simulations used the SI index to understand in "which" fitness landscape the population

was. I think this is a very cool idea. However, this should be a bit more explained to make it clear what was done. Or maybe add to figure 1 an explanation of what was done?

L262:263 - I suggest putting this in an affirmative sentence instead of a question

L265- Instead of continuous model it would be better shifting landscapes (or dynamic) model (as is mentioned above)

L455: Here T is 1(unit of 4-hour competition) or 4hours? From Matuszewski et al 2016, it could also be hours, and then this way it would better reflect the differences in generation time that exists between wildtype and mutated populations.

L468:469 - Was this distribution normal? typically fitness effects are exponentially distributed. Please add the DFE as a supplementary figure.

L500 - comparator —> maybe control or reference?

L506: 508 - Is this a usual metric? if so, can it be referenced? otherwise please clarify why this metric was chosen

L517: 519 - This sentence is not clear. Please rephrase or break the sentence in two?

L528: Please specify the mutation rate used

L547:554 - Why did you use these specific frequencies? why not the real population frequencies that were measured for each fitness landscape?

Supplementary figure 6 and 7 seem to be the same figure (and have the same legend, except for the title). Can you please specify the differences between them?

Reviewer #2:

Remarks to the Author:

The authors study the evolutionary diversification of bacteriophage lambda where genotypes specialising on one or the other of two receptors, LamB and OmpF, evolve in a single experiment. Their key finding is that the evolution is governed by fitness landscapes that deform as new genotypes evolve, in line with the theory of adaptive dynamics. They buttress the conclusions through numerical simulations which show that evolution and long-term coexistence of heterogeneity is possible in their system only with changing landscapes.

The paper addresses important questions of broad interest in evolutionary theory by providing direct experimental evidence for key ideas arising from the theory of adaptive evolution on frequency-dependent fitness landscapes. The methodology of the work appears sound and the robustness of the results are convincingly demonstrated. The presentation of the material is clear and detailed, except for a few points that need to be addressed (see List of questions and suggestions). The paper represents a significant advance in our understanding of evolutionary diversification.

List of questions and suggestions

Major points

1. For the simulations on changing landscapes, the fitness landscape was computed as a weighted

average of the nearest landscapes in terms of SI. I am curious to know why it was done this way. The simplest option seems to be to use only the closest landscape, whereas the other natural option is to use a weighted sum of all the landscapes. Do these choices affect the conclusions about the coexistence of the specialists?

2. I would urge the authors to consider if they can tell us more about the evolution of SI over time in their experiments. This is difficult, but can at least a rough estimate be produced using the mutation frequency data (Supplementary Figure 1) and perhaps combining it with the fitness landscape data (Figure 3). If this is not feasible, it can at least be done numerically. I am talking about a plot similar to Figure 4a, but where the deviation of SI from 0 is shown.

Minor points

1. Line 38: "potentially ubiquitous mechanism". What is the specific mechanism being referred to? Evolution on dynamically deforming landscapes due to the rise of new genotypic subpopulations seems too broad to qualify as a single mechanism.

2. Line 56: "which was previously shown to.." Reference needed.

3. Line 69: "ADT is an extension of game theory applied to evolution of resource-limited populations." This is not quite accurate since ADT is broader in scope, even if some of its better-known applications are in resource-limited populations. Also, I find a bit more reference to ADT and related theoretical literature would be useful.

4. Line 70: "under a typical model of evolution". Could the authors clarify? Do they mean models with frequency-independent selection, perhaps specifically in the strong-selection weak-mutation regime?

5. Figure 1d: Some numbers on the x and y axes would be helpful.

6. Supplementary figure 1: Some uncertainty estimates on the plots would be useful.

7. Figures 1-4: The figure panels are labelled in small letters a, b, c..., but the labels in the caption text are in big letters A, B, C...

8. Figure 2: The rows in the first two cases in Figure 2a (G and CxG) are not aligned with the rows in the remaining cases. This impedes the easy reading of the figure.

9. Figure 3b-d: There seems to be a trend towards decreasing average fitness with increasing mutation number, especially with the L-specialists. Is the reason for this known?

10. Line 229: "Visual inspection of the fitness landscape ... (Figure 3c) shows that an early fitness peak can be reached by an adaptive walk taking only several mutations". This is not easy to see in the existing figure. Can a particular adaptive walk be pointed out with thicker or differently-colored lines in the figure?

11. In Figure 4a, the x-axis is in generations. Can an approximate equivalence be established between generations and time (in days) to facilitate comparison with the experimental data?

12. Line 528: I would suggest quoting the value of mutation rate used for the computation.

Authors' Response to Reviewer Comments

Thank you for handling our submitted manuscript. We have thoroughly reviewed the comments from the two reviewers who have provided detailed suggestions that we have taken into consideration to improve the manuscript. The entirety of the reviewer comments are quoted with beige background below, with our responses in plain text and our quotes from the revised manuscript in grey background. We found many helpful comments from both reviewers that inspired us to perform additional analyses, and we revised many portions of the text and figures to increase clarity and further contextualize our central findings. Please find below a point-by-point summary of our response to these comments.

Reviewer #1 (Remarks to the Author):

In this study the authors study how competition between two virus specialists shapes the fitness landscape. To start they repeat a speciation experiment in lambda virus that is adapting to different host cells. Using mutations identified during evolution to the different host cells they build the fitness landscape for these 9 mutations and test their effect in different environments: presence of different competitors and different host cells. Finally, they simulate evolution using the empirical genotype-phenotype-map with static and dynamic fitness landscapes, that depend on the specialization index of the virus. They show that both the fitness effect of single mutations, as well as their epistatic effect change in response to the presence of competitor and host cell frequency. In addition, they show that considering simulated evolution in dynamic fitness landscapes (instead of static landscapes) allows to better to describe the dynamics observed in the real experiment. This is, to my knowledge, one of the first studies that empirically shows that competition changes fitness landscape dynamically.

This is a very interesting study using a really amazing system to study dynamic fitness landscapes. The results and methods are easy to read and provide a very detailed explanation of the different experiments and combination of simulations and empirical data. Overall, the manuscript is well written. However, there is no discussion of the literature in the study, the results and discussion section although providing a very thorough overview of the results, do not put them into context of what is known of the literature and how the results compare and advance our knowledge. This is also seen in the introduction. Thus, most of my comments are about what literature is missing in the different sections to answer the questions posed by the authors. I focused on the discussion, because it is the easiest to comment, but then the introduction should be updated accordingly. Comments are provided in order of appearance.

We are appreciative of these comments from Reviewer #1. We have thoroughly revised the text as detailed below. In the process, we have added 14 additional references not cited in the original text. Most of these are among the suggested literature listed by Reviewer #1 below. Some of the listed suggestion references were already cited in the original text.

Introduction - The introduction needs a restructure. The idea is super nice, but it is only understood in the final part of the introduction. Maybe it would make the intro flow better to start by explaining fitness landscapes and what is thought to happen during adaptation/speciation. Then introduce the adaptive dynamics and talk about previous work? There is very little information about all the work that has been done in viruses and fitness landscape. Finally, although the study system is important, this should be done in a smaller paragraph together with the specific questions of the study.

We appreciate the reviewer's assessment of the strengths of the paper and their constructive critiques. We

have revised the text to restructure the style of the introduction, expand the discussion of the literature, and better contextualize the main findings in this work. We also shortened a portion of the introduction relating to the study system.

L30: 44 - The first two sections of the introduction seem to be trying to strongly motivate why virus are interesting and can be used to study this topic. However, typically the introduction starts with the general picture of the main topic of the study (e.g. competition driving evolution or fitness landscapes) and its relevance should be presented here. The motivation to use virus can be introduced later.

We have thoroughly revised the introduction in response to this specific comment and the general critiques above. Specifically, we restructured the order of topics in the introduction to first introduce fitness landscapes, followed by a brief introduction to adaptive dynamics and its predictions, and then we use a shortened introduction to the model system we used to study how competition can drive shifts in fitness landscapes and what the implications are on diversification and speciation.

L68: what key prediction?

The key prediction is now more clearly defined in the revised version:

One of the theory's key predictions is that as populations adapt and compete, there are resulting deformations in their fitness landscapes that promote diversification and speciation.

L76:79 - This sentence is complex and should be rephrased. Maybe breaking it in two, would make it clearer?

Thank you for this feedback. We broke this sentence into two sentences for clarity:

ADT predicts that this eco-evolutionary feedback can deform the fitness landscape to allow new populations to ascend new fitness peaks. The original population remains well-adapted to the original peak as the new population emerges, allowing for ecological diversification and speciation.

L81 - remove phenotype from the sentence.

We revised this sentence to read:

Bono et al. used experimental evolution of a bacteriophage to show that the rate of acquiring the ability to infect a new host was associated with the degree of competition between viruses to infect host cells⁷.

L100: 101 - "We repeated the experiment to ensure repeatability in a new location and to preserve samples every 40 hours to track evolutionary dynamics." - The end of this sentence does not make sense, maybe there is a verb missing?

We revised this for clarity:

We repeated the experiment to ensure repeatability in a new location and to preserve samples every 40 hours for longitudinal analyses.

L101: J gene

Revised, thank you.

L107: "All nine mutations resulted in nonsynonymous changes" - Later on, mutations 2 and 3 are coupled together because they introduce the same amino acid change (together or alone). So at least these 2 mutations are synonymous. Is this correct?

For clarification, the most important distinction when asking if the mutations are synonymous is to ask, *synonymous with what*, as we generally are writing about the effects of mutations *relative to the ancestral sequence*. Mutations 2 and 3 are within the same codon, which encodes serine in the ancestral sequence. Mutations 2 and 3 both in isolation and in combination result in a mutant codon encoding arginine. So it is *not* correct to say that mutations 2 and/or 3 are synonymous with respect to the ancestral sequence - they are nonsynonymous with the ancestral sequence. However, it is correct - as the reviewer notes - that

mutations 2 and 3 are synonymous with respect to each other. To increase clarity around this distinction, we revised the caption for Fig. 1a to include the following text:

Some mutations are in the same codon: mutations 2 and 3 (in isolation or in combination) introduce the same nonsynonymous change at that codon

L110 - This title is uninformative (in comparison with the other titles), maybe a small title summarizing the results from this section would be better, for example: High repeatability of the fitness landscape that leads to virus speciation?

We rewrote this title as “**Reproducible measurements of the fitness landscape that leads to speciation**”.

L 145 - The selective coefficient was calculated in one unit of time, but this corresponds to around two generations. Why not use the expected number of generations of the wild type instead of assuming one time unit? Alternatively, using the number of hours. This allows to scale better the effect of the mutant in relation to the wild type. See also: Matuszewski, S., Hildebrandt, M. E., Ghenu, A. H., Jensen, J. D., & Bank, C. (2016). A statistical guide to the design of deep mutational scanning experiments. *Genetics*, 204(1), 77-87.

The expected number of generations is not a precise estimate and is provided in the text to afford the reader a general qualitative gestalt of the experimental system. In response to this specific comment, as well as a related comment by Reviewer #2 below, we have expanded the text to better convey the uncertainties in generation times:

The generation time of λ in our experiments is difficult to know precisely. Generation time depends on adsorption rate (which itself depends on J genotype¹⁴ and host and phage densities during the course of the experiment) and on lysis time, which exhibits some degree of variability⁴².

We sought to minimize the number of assumptions that would need to be made in the process of analyzing the data. For instance, due to the complex ecology (and changing ecology, as we found) of the competition experiments, it is not clear if we can make the assumption that the adsorption time remains constant across genotypes and across the time period of the experiment. We prefer to use a standardized unit of time T (representing a four hour competition experiment) which is consistent across all experiments.

L154: 174 - This section is only results, but there are already several papers that tackle how epistasis is environment dependent and can be change by presence of competitors. Some references that may be interesting to look at for this:

- Hall, A. E., Karkare, K., Cooper, V. S., Bank, C., Cooper, T. F., & Moore, F. B. G. (2019). Environment changes epistasis to alter trade-offs along alternative evolutionary paths. *Evolution*, 73(10), 2094-2105.- Baier et al 2023 Science
- Lalić, J., & Elena, S. F. (2013). Epistasis between mutations is host-dependent for an RNA virus. *Biology letters*, 9(1), 20120396.
- Flynn, K. M., Cooper, T. F., Moore, F. B., & Cooper, V. S. (2013). The environment affects epistatic interactions to alter the topology of an empirical fitness landscape. *PLoS genetics*, 9(4), e1003426.
- Bank, C. (2022). Epistasis and adaptation on fitness landscapes. *Annual review of ecology, evolution, and systematics*, 53, 457-479.
- Cervera, H., Lalić, J., & Elena, S. F. (2016). Effect of host species on topography of the fitness landscape for a plant RNA virus. *Journal of Virology*, 90(22), 10160-10169.

Thank you for providing an extensive list of references in this specific comment and the many other detailed comments below. We have made many revisions throughout the manuscript to better represent the most pertinent literature around the questions discussed in our work, and to contextualize our findings within the

field. For example, in response to this specific comment, an excerpt of this section of the manuscript now reads:

The finding that epistatic interactions between mutations can vary across competitive contexts is in line with broader work demonstrating more generally how environmental perturbations can lead to shifts in fitness landscapes through epistasis, such as the effects of magnesium ion concentration on ribozyme activity²³, nutritional resources for bacterial growth^{10,12}, host species for viral infection^{15,24}, and inducer concentrations regulating a synthetic gene regulatory network¹³.

L155- The initial sentence of the paragraph should introduce the reader to why is it important to study the effect of epistasis. Currently, it reads more as a title of a sub-section than an initial sentence.

Revised as:

It remains an open question to what degree environmental change (such as the spectrum of competitive environments we used to measure fitness landscapes) can reshape fitness landscapes through epistasis².

L175: 186 – In this section it would be important to define specialists and generalists. There is already a plethora of literature that talks about this. A good place to start would be to see:

- Elena SF, Agudelo-Romero P, Lalić J. The evolution of viruses in multi-host fitness landscapes. *Open Virol J.* 2009 Mar 19;3:1-6. doi: 10.2174/1874357900903010001. PMID: 19572052; PMCID: PMC2703199.
- Kassen, R. (2002). The experimental evolution of specialists, generalists, and the maintenance of diversity. *Journal of evolutionary biology*, 15(2), 173-190.
- Syller, J., & Grupa, A. (2016). Antagonistic within-host interactions between plant viruses: molecular basis and impact on viral and host fitness. *Molecular plant pathology*, 17(5), 769-782.

We now define generalist and specialist:

Broadly speaking, receptor ‘generalists’ are defined to utilize more than one receptor, while ‘specialists’ predominantly utilize a single receptor.

We also relate our findings to those of Elena *et al.*:

This is consistent with prior work suggesting that mutations allowing a generalist to specialize on one host are generally accompanied by a trade-off in fitness on other hosts²⁵.

L203: 206 - Here makes sense to discuss DMI in respect to what is known in the literature. Are they prevalent overall? is this common in virus? Some references that may be useful:

- Duffy S, Burch CL, Turner PE. Evolution of host specificity drives reproductive isolation among RNA viruses. *Evolution.* 2007 Nov;61(11):2614-22. doi: 10.1111/j.1558-5646.2007.00226.x. Epub 2007 Oct 1. PMID: 17908251; PMCID: PMC7202233.
- Zhao L, Seth-Pasricha M, Stemate D, Crespo-Bellido A, Gagnon J, Draghi J, Duffy S. Existing Host Range Mutations Constrain Further Emergence of RNA Viruses. *J Virol.* 2019 Feb 5;93(4):e01385-18. doi: 10.1128/JVI.01385-18. PMID: 30463962; PMCID: PMC6364021.
- Paixão, T., Bassler, K. E., & Azevedo, R. B. (2014). Emergent speciation by multiple Dobzhansky–Muller incompatibilities. *bioRxiv*, 008268.
- Unckless RL, Orr HA. Dobzhansky-Muller incompatibilities and adaptation to a shared environment. *Heredity (Edinb).* 2009 Mar;102(3):214-7. doi: 10.1038/hdy.2008.129. Epub 2009 Jan 14. PMID: 19142201; PMCID: PMC2656211.

We have updated the following discussion about genetic incompatibilities to contextualize within the literature:

Overall, the widespread observation of genetic incompatibilities between mutations observed in different receptor specialists shows that Mueller-Dobzanski incompatibilities²⁶, which have not been extensively studied in viral speciation, can contribute to λ 's speciation, in addition to the reproductive isolation ensuing from the preference to infect different host cells^{27,28}.

L223: 225 - How does this compare what is described in the literature? is this common? There aren't many studies, especially in viruses, but maybe looking to other systems would also be important. Some references:

- Martin, C. H., & Wainwright, P. C. (2013). Multiple fitness peaks on the adaptive landscape drive adaptive radiation in the wild. *Science*, 339(6116), 208-211.
- Rainey, P. B., & Travisano, M. (1998). Adaptive radiation in a heterogeneous environment. *Nature*, 394(6688), 69-72.
- Hendry, A. P., Nosil, P., & Rieseberg, L. H. (2007). The speed of ecological speciation. *Functional ecology*, 21(3), 455.

We found the work of Martin et al (in the 2013 citation provided by Reviewer #1 here as well as several follow-up works) particularly interesting and related to our findings, and appreciate Reviewer #1 pointing this out. We discuss our findings with respect to those studies in the revised manuscript by including the following addition to the discussion section of the manuscript:

To our knowledge this is the first measurement of the effect that competitor viruses have on reshaping fitness landscapes and the positions of generalists and specialists on different fitness peaks. Martin *et al.* observed complex fitness landscapes of *Cyprinodon* pupfishes, where a generalist species occupied a local fitness optimum separated by fitness valleys from even higher fitness peaks containing specialists²⁹. They hypothesized competition may allow for the generalists to 'escape' from being trapped on the generalist peak, although intriguing follow-up work found that these fitness peaks were surprisingly static and independent of competitor frequency³⁰. This is in contrast to our findings with λ where there are clearly competitor-dependent shifts in specialist fitness peaks, which are generally accessible to the generalist without crossing a fitness valley. The differing results in these studies on *Cyprinodon* and λ might be due to any number of factors, including differences in the biology and ecology between pupfish and their prey in natural environments, and viruses and their prey in controlled laboratory environments, as well as differences in methodology in measuring fitness landscapes between these studies.

L226: 292 – This final section, should wrap up the different predictions done by dynamic/shifting landscapes and static landscapes. What is known about fitness landscapes when other species evolve. The few work that has been developed are mostly about host-parasite coevolution, but still it would be important to discuss what are the differences as similarities and put them into context. Some references that can be useful:

- Amado, A., & Bank, C. (2023). Ecological tradeoffs lead to complex evolutionary trajectories and sustained diversity on dynamic fitness landscapes. *bioRxiv*, 2023-10.
- Peri, G., Gibard, C., Shults, N. H., Crossin, K., & Hayden, E. J. (2022). Dynamic RNA fitness landscapes of a group I ribozyme during changes to the experimental environment. *Molecular biology and evolution*, 39(3), msab373.
- Rubin, I. N., Ispolatov, Y., & Doebeli, M. (2023). Adaptive diversification and niche packing on rugged fitness landscapes. *Journal of Theoretical Biology*, 562, 111421.
- Bajić, D., Vila, J. C., Blount, Z. D., & Sánchez, A. (2018). On the deformability of an empirical fitness landscape by microbial evolution. *Proceedings of the National Academy of Sciences*, 115(44), 11286-11291.

- Martin, C.H. and Gould, K.J. (2020), Surprising spatiotemporal stability of a multi-peak fitness landscape revealed by independent field experiments measuring hybrid fitness. *Evolution Letters*, 4: 530-544. <https://doi.org/10.1002/evl3.195>
- Patton, A. H., Richards, E. J., Gould, K. J., Buie, L. K., & Martin, C. H. (2022). Hybridization alters the shape of the genotypic fitness landscape, increasing access to novel fitness peaks during adaptive radiation. *Elife*, 11, e72905.
- Gavrilets, Sergey, ‘High-Dimensional Fitness Landscapes and Speciation’, in Massimo Pigliucci, and Gerd B. Müller (eds), *Evolution—the Extended Synthesis* (Cambridge, MA, 2010; online edn, MIT Press Scholarship Online, 22 Aug. 2013), <https://doi.org/10.7551/mitpress/9780262513678.003.0003>,
- Braga, M. P., Araujo, S. B., Agosta, S., Brooks, D., Hoberg, E., Nylin, S., . . . & Boeger, W. A. (2018). Host use dynamics in a heterogeneous fitness landscape generates oscillations in host range and diversification. *Evolution*, 72(9), 1773-1783.
- Williams, H.T. Phage-induced diversification improves host evolvability. *BMC Evol Biol* 13, 17 (2013). <https://doi.org/10.1186/1471-2148-13-17>
- Aguirre, J., Catalán, P., Cuesta, J. A., & Manrubia, S. (2018). On the networked architecture of genotype spaces and its critical effects on molecular evolution. *Open biology*, 8(7), 180069.

Again we find ourselves appreciative of these detailed lists of related literature, and have added discussion around the most pertinent and related of these suggested papers. Specifically, an excerpt now added in the revised manuscript reads:

Our results build upon existing work detailing the effect of eco-evolutionary feedback on adaptive evolution. These results that we obtained using empirical fitness landscapes are in striking agreement with a recent study using theoretical fitness landscapes, where it was also found that dynamic fitness landscapes influenced by eco-evolutionary feedback promoted genetic diversity, whereas static fitness landscapes only allowed for one dominant genotype³¹. Other work examining the deformability of empirical landscapes by metabolic mutations in *E. coli* has suggested that static fitness landscapes can be used to forecast evolution over short evolutionary distances, and only under longer mutational distances do the shifts in fitness landscapes become meaningful³², however our work presents an example where rapid eco-evolutionary feedback can significantly reshape landscapes in a short amount of evolutionary time.

L231: This seems like a weird sentence. Either be specific of how many mutations, or remove the only.

Thank you for pointing this out, we removed “only” from the sentence.

L249: 250 – When I read this section, my main comment was that it is not clear how is it possible to decouple the observations from the expectation? i.e. if the fitness landscape is computed by the weighted average of the population, then of course that it will show that the population is shifting. The genotype-phenotype-fitness map should be fixed. However, when reading the methods this became clearer, so to be sure, you computed the FL from the experimentally obtained data, but assuming a prevalence of competitor of 90% and 10% of a mix and then in the simulations used the SI index to understand in “which” fitness landscape the population was. I think this is a very cool idea. However, this should be a bit more explained to make it clear what was done. Or maybe add to figure 1 an explanation of what was done?

We have added Supplementary Fig. 10 and revised several portions of the text to better describe how the shifting fitness landscape models are implemented in our evolutionary simulation study. These revisions are in response to this specific comment by Reviewer #1, as well as a related comment by Reviewer #2 below which prompted us to test an additional method of implementing shifting of the fitness landscape throughout the simulation. We added the following details to the revised text for clarification:

For each evolved generation in the shifting models, the fitness landscape used to govern selection on that generation was re-assessed based on the simulated population’s average SI. Among the two shifting models we implemented, a fitness landscape was either computed as a weighted average of the nearest two empirical

landscapes (continuous shifting model), or the nearest empirical landscape was used without interpolation (discrete shifting model). In this manner, the fitness landscapes underlying selection in the shifting models fluctuate in response to the emergence of new phenotypes.

L262:263 - I suggest putting this in an affirmative sentence instead of a question

We have revised this sentence to be a statement, instead of posing a question.

L265- Instead of continuous model it would be better shifting landscapes (or dynamic) model (as is mentioned above)

In light of both this specific comment as well as Reviewer #2's main question regarding the implementation of shifting fitness landscape models, we have revised all portions of the text to broadly define our evolutionary models as using *static* vs. *shifting* fitness landscapes. As discussed below there are two *shifting* models reported in the revised text: a *continuous* shifting model which was the model reported in the original submission, and now a *discrete* shifting model additionally described and reported in the revised version of the manuscript.

L455: Here T is 1(unit of 4-hour competition) or 4hours? From Matuszewski et al 2016, it could also be hours, and then this way it would better reflect the differences in generation time that exists between wildtype and mutated populations.

For clarification, in our calculations throughout this work, T is 1. For simplicity, and to avoid making assumptions about the generation time (as elsewhere discussed above and below), we use this scale of time consistently for all calculations. This T of 1 in effect represents a single unit of a 4-hour competition experiment, as noted in the comment.

L468:469 - Was this distribution normal? typically fitness effects are exponentially distributed. Please add the DFE as a supplementary figure.

We have added the distribution of fitness effects for each of the five fitness landscapes in Supplementary Fig. 14. Interestingly, the distributions depend on the competition environment, and mostly appear bimodal in the intermediate competition regimes. This is perhaps reflective of adaptation to two different niches.

L500 - comparator —> maybe control or reference?

We standardized our language throughout the manuscript for this definition to use the word *reference* as opposed to *comparator*.

L506: 508 - Is this a usual metric? if so, can it be referenced? otherwise please clarify why this metric was chosen

We revised the manuscript by adding the following text to better explain our rationale for this approach:

We used a conservative threshold in SI to categorize some genotypes as being receptor specialists for this analysis, since there is no conventional quantitative definition of specialists in this context. We defined receptor specialists as those genotypes with absolute $SI > 0.33$. We chose this cut-off value because based on the formula above for SI, a cut-off value of $SI = 0.33$ defines a specialist genotype as one that has a growth rate approximately two (or more) times greater on the specialized receptor than the non-specialized receptor.

L517: 519 - This sentence is not clear. Please rephrase or break the sentence in two?

Revised and broke into two sentences; it now reads:

The magnitude of this vector measures the change in average receptor fitness in the hybrid genotype relative to the two parental genotypes. Positive and negative values denote improved or worsened average receptor fitness in the hybrid relative to the two parents.

L528: Please specify the mutation rate used

We now include this value in the methods section text.

L547:554 - Why did you use these specific frequencies? why not the real population frequencies that were measured for each fitness landscape?

We used these specific frequencies because they reflected the experimental conditions used to measure the fitness landscapes; in other words (also seen on Fig 1c) we controlled the mixture of the mutant virus library and the competitor strain by mixing previously titrated plaque-forming units of each to achieve the 90% : 10% mixture.

We also revised the text at this position noted by the reviewer for clarity, and it now reads:

We computed the population SI for the experimental conditions used to measure the fitness landscapes as follows: in the three landscapes measured with both L⁻ and O⁻ host cells (Fig. 3b-d), population SI was computed with the assumption that the mutant virus library (which by design was present as 10% of the virus population) has a net SI of 0 (as it is a complex mixture of genotypes sampling both generalists and specialists to varying degrees), and the remainder 90% of the population (by experimental design) was comprised of a competitor genotype with a known SI as computed in Fig. 2b (SI for the L-specialist competitor = +0.778, for the generalist EvoC = 0, and for the O-specialist competitor = -0.99). By calculating the population SI for each of these experimental conditions, we positioned landscape B at population SI = +0.7, landscape C at population SI = 0, and landscape D at population SI = -0.9; as these were the population SI present in the experimental conditions that were used to measure each of these landscapes.

Supplementary figure 6 and 7 seem to be the same figure (and have the same legend, except for the title). Can you please specify the differences between them?

We have added clarifying text to the caption of Supplementary Fig. 6:

This is similar to Supplementary Figure 7 but shows the fitness effects of the isolated mutation 3 (without mutation 2) on the y-axis.

We also added this text to the caption of Supplementary Fig. 7:

This is similar to Supplementary Figure 6 but shows the fitness effects of the combined mutations 2+3 on the y-axis.

Reviewer #2 (Remarks to the Author):

The authors study the evolutionary diversification of bacteriophage lambda where genotypes specialising on one or the other of two receptors, LamB and OmpF, evolve in a single experiment. Their key finding is that the evolution is governed by fitness landscapes that deform as new genotypes evolve, in line with the theory of adaptive dynamics. They buttress the conclusions through numerical simulations which show that evolution and long-term coexistence of heterogeneity is possible in their system only with changing landscapes.

The paper addresses important questions of broad interest in evolutionary theory by providing direct experimental evidence for key ideas arising from the theory of adaptive evolution on frequency-dependent fitness landscapes. The methodology of the work appears sound and the robustness of the results are convincingly demonstrated. The presentation of the material is clear and detailed, except for a few points that need to be addressed (see List of questions and suggestions). The paper represents a significant advance in our understanding of evolutionary diversification.

We are appreciative of Reviewer #2's insightful feedback. We have performed a few additional analyses in response to these constructive critiques, as outlined below, and we appreciate that these suggestions have helped strengthen the manuscript overall.

List of questions and suggestions

Major points

1. For the simulations on changing landscapes, the fitness landscape was computed as a weighted average of the nearest landscapes in terms of SI. I am curious to know why it was done this way. The simplest option seems to be to use only the closest landscape, whereas the other natural option is to use a weighted sum of all the landscapes. Do these choices affect the conclusions about the coexistence of the specialists?

We envisioned that we could approximate how the deformations in the fitness landscape are shaped between any two experimentally-measured landscapes by using linear interpolation, in other words taking a weighted average of two adjacent landscapes. By taking this approximation, we implemented the original shifting landscapes model as a *continuous* interpolation between the two empiric landscapes adjacent to the current population SI. We agree with Reviewer #2 that this method of implementing shifting fitness landscapes is slightly more complex (making an assumption of linear transformation along the landscape axis) than the more intuitive approach of simply using the *closest* landscape and making discrete jumps between empirical landscapes whenever the simulated population SI moves to become closer to a new empirical landscape than the current landscape being used to govern selection. Thus, we implemented a shifting landscape model that made *discrete* jumps between empirical fitness landscapes, jumping to a new landscape if the population SI moved closer to it than the previously used landscape. We tested whether our conclusions about the significance of shifting fitness landscapes were dependent on how we implemented the shifts by comparing the results obtained by both the *continuous* and *discrete* shifting models. The main results of these additional analyses are found in a **revised Supplementary Fig. 5** (adding example population abundance curves for the discrete shifting model) and in **Supplementary Fig. 11** (a new figure that reprises the results shown in Figure 4 alongside the new results using the discrete shifting model).

We also added the following to the results section to describe the rationale and results for these additional analyses:

To test whether the results of the continuous shifting model were dependent on the method of continuous interpolation between experimentally-measured landscapes, we also implemented a ‘discrete’ shifting model. The discrete model only used the five experimentally measured landscapes and shifted between landscapes based on which landscape was ‘closest’ on the landscape axis to that generation’s population SI. Only the continuous model is shown in Fig. 4 for brevity; the effect on maintaining genetic and phenotypic diversity is the same in both shifting models (Supplementary Fig. 11).

2. I would urge the authors to consider if they can tell us more about the evolution of SI over time in their experiments. This is difficult, but can at least a rough estimate be produced using the mutation frequency data (Supplementary Figure 1) and perhaps combining it with the fitness landscape data (Figure 3). If this is not feasible, it can at least be done numerically. I am talking about a plot similar to Figure 4a, but where the deviation of SI from 0 is shown.

We really appreciate this idea, which we have implemented to help understand patterns of the evolution of the population SI in our simulation experiments. As the reviewer suggested, the trajectory of population SI can be monitored throughout each simulation and summarized across the various evolutionary models implementing static and shifting fitness landscapes. We made figures showing the population SI over time in the simulations; Supplementary Fig. 12 shows the replicate simulations within each model, and Supplementary Fig. 13 shows the median and interquartile range for each model (analogous to Fig 4a).

We comment on these results in the Supplementary Fig. 12 caption:

The population SI trajectory for each simulation is plotted as a semi-transparent line, for all 500 replicate simulations under each landscape model. The shifting landscape models tend to reach an equilibrium around a population SI between -0.9 and -1.0, in contrast to the static models which tend to reach equilibrium SI near +/- 1.0 as they become fixed with single specialist genotypes and phenotypes (as seen in Fig. 4; also see Supplementary Fig. 13 for trajectories of the median SI across simulations within each model). Landscapes A, B, and C all reach an endpoint with low genetic diversity dominated by O-specialists (as seen in Fig. 4, Supplementary Fig. 11), but the time it takes to reach this equilibrium is shortest in

the most extreme competitive environment (landscape A, characterized by the highest fitness peaks of the dominating O-specialists). There is more stochasticity across simulations in static landscapes D and E, which is also reflected in the different specialist endpoints reached as shown in Fig. 4b. Landscape E is unique in that the fitness peaks are defined by both generalists and L-specialists (see Fig. 3e); the evolutionary trajectories of SI most often equilibrate to a very slightly negative SI value corresponding to a dominating generalist, but less often also equilibrate to a relatively positive SI corresponding to a dominating L-specialist.

Unfortunately, we cannot take a similar approach to examining the evolution of SI over time in the evolutionary replay experiment because the short-read shotgun sequencing data does not provide full linkage of mutations into complete genotypes; we only obtained full-length sequencing of J in the two endpoint specialists isolated at the end of the experiment.

Minor points

1. Line 38: “potentially ubiquitous mechanism”. What is the specific mechanism being referred to? Evolution on dynamically deforming landscapes due to the rise of new genotypic subpopulations seems too broad to qualify as a single mechanism.

We agree this phrasing was not ideal. In the process of restructuring the introduction in response to critiques from Reviewer #1, this sentence no longer appears in the revised introduction.

2. Line 56: “which was previously shown to..” Reference needed.

Thank you, the reference has been added in the revised text.

3. Line 69: “ADT is an extension of game theory applied to evolution of resource-limited populations.” This is not quite accurate since ADT is broader in scope, even if some of its better-known applications are in resource-limited populations. Also, I find a bit more reference to ADT and related theoretical literature would be useful.

We have revised the text to be more technically correct, and cited additional pertinent literature in the revised text:

Adaptive dynamics theory (ADT) is a framework for studying evolutionary change in a setting where the fitness of individuals is not static over time, but can be affected by the frequency of the individuals as well as changes in their ecology³⁻⁵. ADT is an extension of game theory^{4,6} and has been applied to study evolution of resource-limited populations, where competition can dynamically influence fitness landscapes.

4. Line 70: “under a typical model of evolution”. Could the authors clarify? Do they mean models with frequency-independent selection, perhaps specifically in the strong-selection weak-mutation regime?

We revised this sentence to emphasize that we mean evolution along a *fixed fitness landscape*.

5. Figure 1d: Some numbers on the x and y axes would be helpful.

The revised figure now retains axis tick labels.

6. Supplementary figure 1: Some uncertainty estimates on the plots would be useful.

Unfortunately it is not clear how to provide uncertainty estimates in retrospect, since the population sequencing for each timepoint was only done as a single measurement. We agree that when possible, uncertainty estimates would be helpful. However, we note that these data are mainly interpreted *qualitatively*; these data simply helped us determine *which* sites and mutations to include in the genotype space surveyed to measure fitness landscapes.

7. Figures 1-4: The figure panels are labelled in small letters a, b, c..., but the labels in the caption text are in big letters A, B, C...

We have revised the formatting of the caption text with lower-case letters to reflect the figure panel labels.

8. Figure 2: The rows in the first two cases in Figure 2a (G and CxG) are not aligned with the rows in the remaining cases. This impedes the easy reading of the figure.

Thank you for pointing this out. We have revised the figure to align the rows across the entire figure panel.

9. Figure 3b-d: There seems to be a trend towards decreasing average fitness with increasing mutation number, especially with the L-specialists. Is the reason for this known?

We do not have a clear explanation for this observation, but presumably this is the product of some sort of protein-level epistasis that affects the mechanism of how the J protein engages the LamB receptor more than the mechanism of how it engages OmpF; these areas are not understood and ripe for further study.

10. Line 229: “Visual inspection of the fitness landscape . . . (Figure 3c) shows that an early fitness peak can be reached by an adaptive walk taking only several mutations”. This is not easy to see in the existing figure. Can a particular adaptive walk be pointed out with thicker or differently-colored lines in the figure?

We revised the figure by adding two small black arrows to panel C. These arrows symbolize two sequential mutations that can be taken from the ancestral sequence to reach the described fitness peak.

11. In Figure 4a, the x-axis is in generations. Can an approximate equivalence be established between generations and time (in days) to facilitate comparison with the experimental data?

As discussed in response to similar questions raised by Reviewer #1, the expected number of generations is not a precise estimate and is provided in the text only to afford the reader a general qualitative gestalt of the experimental system. In the revised manuscript, we discuss the limitations of making assumptions about generation times while also providing a rough estimate under the section **Computer simulations of λ evolution**:

The generation time of λ in our experiments is difficult to know precisely. Generation time depends on adsorption rate (which itself depends on J genotype¹⁴ and host and phage densities during the course of the experiment) and on lysis time, which exhibits some degree of variability⁴². Using an approximation of 60 minutes as the generation time, each eight-hour ‘day’ in the speciation replay experiment would reflect 8 generations, resulting in a total of $8 \times 35 = 280$ generations in the replay experiment. We conservatively chose to run our simulations for 500 generations to ensure that the simulations would be long enough to sufficiently capture evolutionary dynamics occurring in our replay experiment. In most cases across all models, a state of equilibrium is reached by generation 280 and endures until generation 500 (Fig. 4a, Supplementary Fig. 5, Supplementary Fig. 12).

12. Line 528: I would suggest quoting the value of mutation rate used for the computation.

We have added this to the text.

Reviewers' Comments:

Reviewer #1:

Remarks to the Author:

The authors did a great job at answering and clarifying the points raised by the two reviewers. Congratulations on a very cool study and manuscript!

Very few minor comments.

L32: 33 - "and evolving populations can be thought to take adaptive walks across this topography." -- > this sentence reads a bit weirdly because when at first read it seems that populations are purposely walking through the fitness landscape. Maybe just removing the middle part and joining the two sentences would be clearer?

Conceptually, fitness landscapes are often projected as two- dimensional distributions of fitness peaks and valleys, and mutations allow populations to move randomly across the topography. If the individual climbs higher up a fitness peak, survival becomes more likely, and less likely during descent into fitness valleys.

L113 - Refer here to Figure 1c

L150: 151 - "where the most positive and negative coefficients denote the most positive and negative impacts on predicted fitness." - This sentence seems a bit redundant, maybe: where the most positive and negative coefficients denote the strongest impacts on predicted fitness.

Figure 3 the plots are very small, and it is not possible to see the points properly. To improve this, the authors could make a panel with three rows and two columns, to increase the size of each sub-figure so there is higher distance between the points.

Reviewer #2:

Remarks to the Author:

All the issues have been adequately addressed by the authors.

Authors' Response to Reviewer Comments (Revision #2)

REVIEWERS' COMMENTS

Reviewer #1 (Remarks to the Author):

The authors did a great job at answering and clarifying the points raised by the two reviewers. Congratulations on a very cool study and manuscript!

Very few minor comments.

L32: 33 - "and evolving populations can be thought to take adaptive walks across this topography." → this sentence reads a bit weirdly because when at first read it seems that populations are purposely walking through the fitness landscape. Maybe just removing the middle part and joining the two sentences would be clearer? Conceptually, fitness landscapes are often projected as two-dimensional distributions of fitness peaks and valleys, and mutations allow populations to move randomly across the topography. If the individual climbs higher up a fitness peak, survival becomes more likely, and less likely during descent into fitness valleys.

We agree that the phrasing we used is overly anthropomorphic. We revised the sentence to read very similarly to the suggested edit:

Conceptually, fitness landscapes are often projected as two-dimensional distributions of fitness peaks and valleys, and mutations allow populations to move randomly across the topography. If mutations reposition an individual higher up a fitness peak, survival becomes more likely, and conversely survival is less likely during descent into fitness valleys.

L113 – Refer here to Figure 1c

Reference to Figure 1c has been added in the text.

L150: 151 – "where the most positive and negative coefficients denote the most positive and negative impacts on predicted fitness." – This sentence seems a bit redundant, maybe: where the most positive and negative coefficients denote the strongest impacts on predicted fitness.

We have reworded this sentence as suggested by the reviewer.

Figure 3 the plots are very small, and it is not possible to see the points properly. To improve this, the authors could make a panel with three rows and two columns, to increase the size of each sub-figure so there is higher distance between the points.

We acknowledge that it is not possible to see every single point on the fitness landscape figure, and we have considered several ways to improve the readability of this figure in response to this comment. We acknowledge the points are very small, but note that the figure is at relatively high resolution permitting zooming in as desired. Making the points themselves smaller increases the distance between the points, but makes it harder to see the specialization index color mapped to each genotype. Enlarging the subplots by splitting the figure panels into multiple rows would allow for larger plots, but comes at the expense of removing the reader's ability to see all five landscapes on a single "landscape axis" defined by relative competition and availability of the two host receptors, as elaborated in Results and discussion under "Dynamic fitness landscapes promote genotypic and phenotypic diversification", Methods under "Computer simulations of evolution", and Supplementary Fig. 10. Since the specialization index color mapping and the arrangement of

fitness landscapes along an axis are central concepts to the theme of the paper, we have opted to keep the positioning and point sizes unchanged in Figure 3. For added clarity, we have also made a version of Figure 3 with the changes requested by the reviewer (smaller point sizes and larger figure panels to increase the size of each sub-figure and increased distance between points to show as many individual points as possible) - this is now included as Supplementary Fig. 15.

Reviewer #1 (Remarks on code availability):

I have not reviewed the code nor ran it, but I read the supporting information and it seems to have all the information required to re run the analysis, including the anaconda environment information needed.

Reviewer #2 (Remarks to the Author):

All the issues have been adequately addressed by the authors.

Thank you again to both reviewers for the time you took to provide thorough and thoughtful reviews of the original and revised versions of the manuscript. Your reviews certainly helped us greatly improve the manuscript.